# ABC: Auxiliary Balanced Classifier for Class-Imbalanced Semi-Supervised Learning

**Hyuck Lee   Seungjae Shin   Heeyoung Kim**
Department of Industrial and Systems Engineering, KAIST
{dlgur0921, tmdwo0910, heeyoungkim}@kaist.ac.kr

## Abstract

Existing semi-supervised learning (SSL) algorithms typically assume class-balanced datasets, although the class distributions of many real-world datasets are imbalanced. In general, classifiers trained on a class-imbalanced dataset are biased toward the majority classes. This issue becomes more problematic for SSL algorithms because they utilize the biased prediction of unlabeled data for training. However, traditional class-imbalanced learning techniques, which are designed for labeled data, cannot be readily combined with SSL algorithms. We propose a scalable class-imbalanced SSL algorithm that can effectively use unlabeled data, while mitigating class imbalance by introducing an auxiliary balanced classifier (ABC) of a single layer, which is attached to a representation layer of an existing SSL algorithm. The ABC is trained with a class-balanced loss of a minibatch, while using high-quality representations learned from all data points in the minibatch using the backbone SSL algorithm to avoid overfitting and information loss. Moreover, we use consistency regularization, a recent SSL technique for utilizing unlabeled data in a modified way, to train the ABC to be balanced among the classes by selecting unlabeled data with the same probability for each class. The proposed algorithm achieves state-of-the-art performance in various class-imbalanced SSL experiments using four benchmark datasets.

## 1   Introduction

Recently, numerous deep neural network (DNN)-based semi-supervised learning (SSL) algorithms have been proposed to improve the performance of DNNs by utilizing unlabeled data when only a small amount of labeled data is available. These algorithms have shown effective performance in various tasks. However, most existing SSL algorithms assume class-balanced datasets, whereas the class distributions of many real-world datasets are imbalanced. It is well known that classifiers trained on class-imbalanced data tend to be biased toward the majority classes. This issue can be more problematic for SSL algorithms that use predicted labels of unlabeled data for their training, because the labels predicted by the algorithm trained on class-imbalanced data become even more severely imbalanced [18]. For example, Figure 1 (b) presents biased predictions of ReMixMatch [3], a recent SSL algorithm, trained on CIFAR-10-LT, which is a class-imbalanced dataset with the amount of Class 0 being 100 times more than that of Class 9, as depicted in Figure 1 (a). Although there are various class-imbalanced learning techniques, they are usually designed for labeled data, and thus cannot be simply combined with SSL algorithms under class-imbalanced SSL (CISSL) scenarios. Recently, a few CISSL algorithms have been proposed, but the CISSL problem is still underexplored.

We propose a new CISSL algorithm that can effectively use unlabeled data, while mitigating class imbalance by using an existing DNN-based SSL algorithm [3, 29] as the backbone and introducing an auxiliary balanced classifier (ABC) of a single layer. The ABC is attached to a representation

35th Conference on Neural Information Processing Systems (NeurIPS 2021).

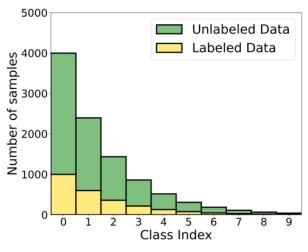
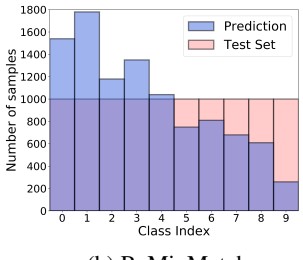
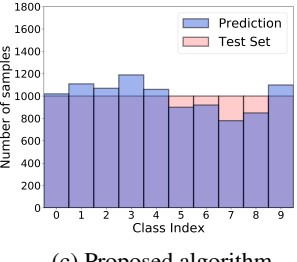

| (a) Class-imbalanced training set | (b) ReMixMatch | (c) Proposed algorithm |

Figure 1: Predictions on a class-balanced test set using ReMixMatch (b) and the proposed algorithm (c) trained on a class-imbalanced training set (a).

layer immediately preceding the classification layer of the backbone, based on the argument that a classification algorithm (i.e., backbone) can learn high-quality representations even if its classifier is biased toward the majority classes [17]. The ABC is trained to be balanced across all classes by using a mask that rebalances the class distribution, similar to re-sampling in previous SSL studies [2, 7, 13, 16]. Specifically, the mask stochastically regenerates a class-balanced subset of a minibatch on which the ABC is trained. The ABC is trained simultaneously with the backbone, so that the ABC can use high-quality representations learned from all data points in the minibatch using the backbone. In this way, the ABC can overcome the limitations of the previous resampling techniques, overfitting on minority-class data or loss of information on majority-class data [6, 27].

Moreover, to place decision boundaries in low-density regions by utilizing unlabeled data, we use consistency regularization, a recent SSL technique, which enforces the classification outputs of two augmented or perturbed versions of the same unlabeled example to remain unchanged. In particular, we encourage the ABC to be balanced across classes when using consistency regularization by selecting unlabeled examples with the same probability for each class using a mask. Figure 1 (c) illustrates that compared to the results of ReMixMatch in Figure 1 (b), the class distribution of the predicted labels becomes more balanced using the proposed algorithm trained on the same dataset. Our experimental results under various scenarios demonstrate that the proposed algorithm achieves state-of-the-art performance. Through qualitative analysis and an ablation study, we further investigate the contribution of each component of the proposed algorithm. The code for the proposed algorithm is available at https://github.com/LeeHyuck/ABC.

## 2 Related Work

**Semi-supervised learning (SSL)** Recently, several SSL techniques that utilize unlabeled data have been proposed. Entropy minimization [12] encourages the classifier outputs to have low entropy for unlabeled data, as in pseudo-labels [22]. Mixup regularization [4, 32] makes the decision boundaries farther away from the data clusters by encouraging the prediction for an interpolation of two inputs to be the same as the interpolation of the prediction for each input. Consistency regularization [26, 24, 30] encourages a classifier to produce similar predictions for perturbed versions of the same unlabeled input. To create perturbed unlabeled inputs, various data augmentation techniques have been used. For example, FixMatch [29] and ReMixMatch [3] used strong augmentation methods such as Cutout [10] and RandomAugment [8]. FixMatch and ReMixMatch are used as the backbone of the proposed algorithm; they are described in Section 3.2.

**Class-imbalanced learning (CIL)** As a popular approach for CIL, re-sampling techniques [16, 7, 2, 13] balance the number of training samples for each class in the training set. As another popular approach, re-weighting techniques [23, 14, 33] re-weight the loss for each class by a factor inversely proportional to the number of data points belonging to that class. Although these approaches are simple, they have some drawbacks. For example, oversampling from minority classes can cause overfitting, whereas undersampling from majority classes can cause information loss [6]. In the case of re-weighting, gradients can be calculated to be abnormally large when the class imbalance is severe, resulting in unstable training [6, 1]. Many attempts have been made to alleviate these problems, such as effective re-weighting [9] and meta-learning-based re-weighting [28, 15]. New forms of losses have also been proposed [6, 27]. In [36, 19], knowledge is transferred from the data of

majority classes to the data of minority classes. These CIL algorithms were designed for labeled data and require label information; thus, they are not applicable to unlabeled data. In [17], it was found that biased classification is mainly due to the classification layer and that a classification algorithm can learn meaningful representations even from a class-imbalanced training set. Based on this finding, we design the ABC to use high-quality representations learned from class-imbalanced data utilizing FixMatch [29] and ReMixMatch [3].

**Class-imbalanced semi-supervised learning (CISSL)** There have been few studies on CISSL. In [35], it was found that more accurate decision boundaries can be obtained in class-imbalanced settings through self-supervised learning and semi-supervised learning. DARP [18] refines biased pseudo-labels by solving a convex optimization problem. CReST [34], a recent self-training technique, mitigates class imbalance by using pseudo-labeled unlabeled data points classified as minority classes with a higher probability than those classified as majority classes.

## 3 Methodology

### 3.1 Problem setting

Suppose that we have a labeled dataset $\mathcal{X} = \{(x_n, y_n) : n \in (1, ..., N)\}$, where $x_n \in \mathbb{R}^d$ is the $n$th labeled data point and $y_n \in \{1, ..., L\}$ is the corresponding label. We also have an unlabeled dataset $\mathcal{U} = \{(u_m) : m \in (1, ..., M)\}$, where $u_m \in \mathbb{R}^d$ is the $m$th unlabeled data point. We express the ratio of the amount of labeled data as $\beta = \frac{N}{M+N}$. Generally, $\beta < 0.5$, because label acquisition is costly and laborious. We denote the number of labeled data points of class $l$ as $N_l$, i.e., $\sum_{l=1}^{L} N_l = N$, and assume that the $L$ classes are sorted according to cardinality in descending order, i.e., $N_1 \geq N_2 \geq \cdots \geq N_L$. We denote the ratio of the class imbalance as $\gamma = \frac{N_1}{N_L}$. Under class-imbalanced scenarios, $\gamma \gg 1$. Following previous CIL studies, we define the half of the classes containing a large amount of data as the majority classes, and the other half of the classes, containing a small amount of data, as the minority classes. Following [34], we assume that $\mathcal{X}$ and $\mathcal{U}$ share the same class distribution, i.e., the labeled and unlabeled datasets are class-imbalanced to the same extent. From $\mathcal{X}$ and $\mathcal{U}$, we generate minibatches $\mathcal{MB}_{\mathcal{X}} = \{(x_b, y_b) : b \in (1, ..., B)\} \subset \mathcal{X}$ and $\mathcal{MB}_{\mathcal{U}} = \{(u_b) : b \in (1, ..., B)\} \subset \mathcal{U}$ for each iteration of training, where $B$ is the minibatch size. Using these minibatches for training, we aim to learn a model $f : \mathbb{R}^d \to \{1, ...L\}$ that performs effectively on a class-balanced test set.

### 3.2 Backbone SSL algorithm

We attach the ABC to the backbone's representation layer, so that it can utilize the high-quality representations learned by the backbone. We use FixMatch [29] or ReMixMatch [3] as the backbone, as these two have achieved state-of-the-art SSL performance. FixMatch uses the classification loss calculated from the weakly augmented labeled data point $\alpha(x_b)$ generated by flipping and cropping the image, and the consistency regularization loss calculated from the weakly augmented unlabeled data point $\alpha(u_b)$ and strongly augmented unlabeled data point $\mathcal{A}(u_b)$ generated by Cutout [10] and RandomAugment [8]. ReMixMatch predicts the class label of the weakly augmented unlabeled data point $\alpha(u_b)$ using distribution alignment and sharpening, and assigns the predicted label to the strongly augmented unlabeled data point $\mathcal{A}(u_b)$. These strongly augmented unlabeled data point $\mathcal{A}(u_b)$ and strongly augmented labeled data point $\mathcal{A}(x_b)$ are used to conduct mixup regularization. ReMixMatch also conducts consistency regularization in a manner similar to FixMatch and self-supervised learning using the rotation of the image [11, 39]. FixMatch and ReMixMatch have greatly improved the SSL performance by learning high-quality representations using strong data augmentation. However, these algorithms can be significantly biased toward the majority classes in class-imbalanced settings.

Using FixMatch and ReMixMatch as the backbone of the proposed algorithm, we ensure that the ABC enjoys high-quality representations learned by the backbone, while replacing the backbone's biased classifier. To train the ABC, we reuse the weakly augmented data and strongly augmented data used by the backbone to decrease the computational cost. Although we use FixMatch and ReMixMatch as the backbone in this study, the ABC can also be combined with other DNN-based SSL algorithms, as long as they use weakly augmented data and strongly augmented data.

### 3.3 ABC for class-imbalanced Semi-supervised learning

To train the ABC to be balanced, we first generate $0/1$ mask $M(x_b)$ for each labeled data point $x_b$ using a Bernoulli distribution $\mathcal{B}(\cdot)$ with the parameter set to be inversely proportional to the number of data points of each class. This setting makes $\mathcal{B}(\cdot)$ generate mask 1 with high probability for the data points in the minority classes, but with low probability for those in the majority classes. Then, the classification loss is multiplied by the generated mask, so that the ABC can be trained with a balanced classification loss. Multiplying the classification loss by the $0/1$ mask can be interpreted as oversampling of the data points in the minority classes, whereas it can be interpreted as undersampling of those in the majority classes. In representation learning, oversampling and undersampling techniques have shown overfitting and information loss problems, respectively. In contrast, the ABC can overcome these problems because it uses the representations learned by the backbone, which is trained on all data points in the minibatch. The use of the $0/1$ mask to construct the balanced loss, instead of directly creating a balanced subset, allows the backbone and the ABC to be trained from the same minibatches. Therefore, the representations of minibatches calculated for training the backbone can be used again for training the ABC. Consequently, the proposed algorithm only requires a slightly increased time cost compared to training the backbone alone. This is confirmed in Section 4.3. The overall procedure of balanced training with $0/1$ mask for the ABC attached to a representation layer of the backbone is presented in Figure 2. The classification loss for the ABC, $L_{cls}$, with $0/1$ mask $M(\cdot)$ is expressed as

$$L_{cls} = \frac{1}{B} \sum_{b=1}^{B} M(x_b) \, \mathbf{H}\left(p_s\left(y|\alpha\left(x_b\right)\right), p_b\right), \tag{1}$$

$$M(x_b) = \mathcal{B}\left(\frac{N_L}{N_{y_b}}\right), \tag{2}$$

where $\mathbf{H}$ is the standard cross-entropy loss, $\alpha(x_b)$ is an augmented labeled data point, $p_s\left(y|\alpha\left(x_b\right)\right)$ is the predicted class distribution using the ABC for $\alpha(x_b)$, and $p_b$ is the one-hot label for $x_b$.

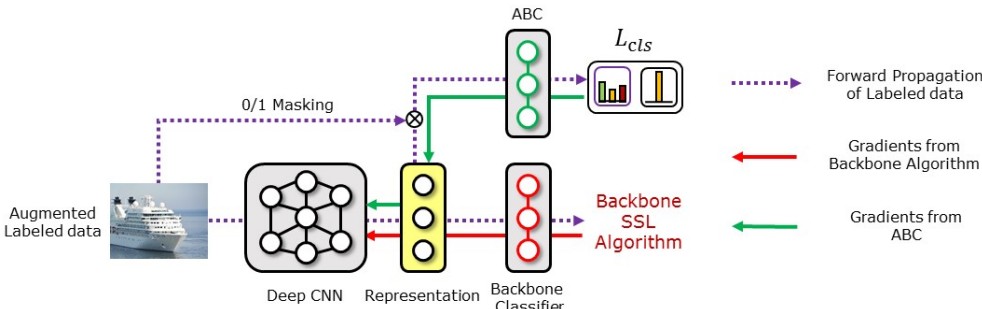

Figure 2: Overall procedure for balanced training of the ABC with a $0/1$ mask

### 3.4 Consistency regularization for ABC

To increase the margin between the decision boundary and the data points using unlabeled data, we conduct consistency regularization for the ABC, similar to the way in FixMatch. Specifically, we first obtain the predicted class distribution $p_s\left(y|\alpha\left(u_b\right)\right)$ for a weakly augmented unlabeled data point $\alpha\left(u_b\right)$ using the ABC and use it as a soft pseudo-label $q_b$. Then, for two strongly augmented unlabeled data points $\mathcal{A}_1\left(u_b\right)$ and $\mathcal{A}_2\left(u_b\right)$, we train the ABC to produce their predicted class distributions, $p_s\left(y|\mathcal{A}_1\left(u_b\right)\right)$ and $p_s\left(y|\mathcal{A}_2\left(u_b\right)\right)$, to be close to $q_b$.

In class-imbalanced settings, because most unlabeled data points belong to majority classes, most weakly augmented unlabeled data points can be predicted as the majority classes. Then, consistency regularization would be conducted with a higher frequency for the majority classes, which can cause a classifier to be biased toward the majority classes. To prevent this issue, we conduct consistency regularization in a modified manner that is suitable for class-imbalance problems. Specifically,

whereas FixMatch minimizes entropy by converting the predicted class distribution for a weakly augmented data point into a one-hot pseudo-label, we directly use the predicted class distribution as a soft pseudo-label. We do not pursue entropy minimization for the ABC because it can accelerate biased classification toward certain classes. Moreover, we once again generate $0/1$ mask $M(\cdot)$ for each unlabeled data point $u_b$ based on a soft pseudo label $q_b$, and multiply the consistency regularization loss for $u_b$ by the generated mask, so that the ABC can be trained with a class-balanced consistency regularization loss. Note that existing resampling techniques are not applicable to unlabeled data, because they require a label for each data point. In contrast, we make it possible to resample unlabeled data by using the soft pseudo-label and the $0/1$ mask. The consistency regularization loss, $L_{con}$, with $0/1$ mask $M(\cdot)$ is expressed as

$$L_{con} = \frac{1}{B} \sum_{b=1}^{B} \sum_{k=1}^{2} M(u_b) \, \mathbf{I}(\max(q_b) \geq \tau) \, \mathbf{H}(p_s(y|\mathcal{A}_k(u_b)), q_b), \tag{3}$$

$$M(u_b) = \mathcal{B}\left(\frac{N_L}{N_{\widehat{q}_b}}\right), \tag{4}$$

where $\mathbf{I}$ is the indicator function, $\max(q_b)$ is the highest predicted assignment probability for any class, representing the confidence of prediction, and $\tau$ is the confidence threshold. To avoid the unwanted effects of inaccurate soft pseudo-labels during consistency regularization, we only use the weakly augmented unlabeled data points whose confidence is higher than the threshold $\tau$, similar to that in FixMatch. To take full advantage of few unlabeled data points with prediction confidence values that are higher than the confidence threshold $\tau$ in the early stage of training, we gradually decrease the parameter of the Bernoulli distribution $\mathcal{B}(\cdot)$ for $u_b$ from 1 to $N_L/N_{\widehat{q}_b}$, where $\widehat{q}_b$ is the one-hot pseudo-label obtained from $q_b$. Following previous studies [3, 24, 29, 4], we do not backpropagate gradients for pseudo-label prediction. The overall procedure for consistency regularization for the ABC is shown in Appendix A.

### 3.5 End-to-end training

Unlike a recent CIL trend to finetune a classifier in a balanced manner after representation learning is completed (i.e., decoupled learning of representations and a classifier) [17, 27], we obtain a balanced classifier by training the proposed algorithm end-to-end. We train the proposed algorithm with the sum of losses from Sections 3.3 and 3.4, and the loss for the backbone, $L_{back}$. The total loss function $L_{total}$ is expressed as

$$L_{total} = L_{cls} + L_{con} + L_{back}. \tag{5}$$

Whereas we use the sum of the losses for the backbone and ABC for training the proposed algorithm, we predict the class labels of new data points using only the ABC. In our experiments in Sections 4.4 and 4.5, we show that the proposed algorithm trained end-to-end produces better performance than competing algorithms with decoupled learning of representations and a classifier, and we analyze possible reasons. We present the pseudo code of the proposed algorithm in Appendix B.

## 4 Experiments

### 4.1 Experimental setup

We created class-imbalanced versions of CIFAR-10, CIFAR-100 [21], and SVHN [25] datasets to conduct experiments under various ratios of class imbalance $\gamma$ and various ratios of the amount of labeled data $\beta$. For class-imbalance types, we first consider long-tailed (LT) imbalance in which the number of data points exponentially decreases from the largest to the smallest class, i.e., $N_k = N_1 \times \gamma^{-\frac{k-1}{L-1}}$, where $\gamma = \frac{N_1}{N_L}$. We also consider step imbalance [5] in which the whole majority classes have the same amount of data and the whole minority classes also have the same amount of data. Two types of class imbalance for the considered datasets are illustrated in Appendix C. For the main setting, we set $\gamma = 100$, $N_1 = 1000$, and $\beta = 20\%$ for CIFAR-10 and SVHN, and $\gamma = 20$, $N_1 = 200$ and $\beta = 40\%$ for CIFAR-100. Similar to [18], we set $\gamma$ of CIFAR-100 to be relatively small because CIFAR-100 has only 500 training data points for each class. To evaluate the

performance of the proposed algorithm on large-scale datasets, we also conducted experiments on 7.5M data points of 256 by 256 images from the LSUN dataset [37].

We compared the performance of the proposed algorithm with that of various baseline algorithms. Specifically, we considered the following baseline algorithms:

- Deep CNN (vanilla algorithm): This is trained on only labeled data with the cross-entropy loss.

- BALMS [27] (CIL algorithm): This state-of-the-art CIL algorithm does not use unlabeled data.

- VAT [24], ReMixMatch [3], and FixMatch [29] (SSL algorithms): These are state-of-the-art SSL algorithms, but do not consider class imbalance.

- FixMatch+CReST+PDA and ReMixMatch+CReST+PDA (CISSL algorithms): CReST+PDA [34] mitigates class imbalance by using unlabeled data points classified as the minority classes with a higher probability than those classified as the majority classes.

- ReMixMatch+DARP and FixMatch+DARP (CISSL algorithms): These algorithms use DARP [18] to refine the pseudo labels obtained from ReMixMatch or FixMatch.

- ReMixMatch+DARP+cRT and FixMatch+DARP+cRT (CISSL algorithms): Compared to ReMix-Match+DARP and FixMatch+DARP, these algorithms finetune the classifier using cRT [17].

For the structure of the deep CNN used in the proposed and baseline algorithms, we used Wide ResNet-28-2 [38]. We trained the proposed algorithm for $250,000$ iterations with a batch size of 64. The confidence threshold $\tau$ was set to $0.95$ based on experiments with various values of $\tau$ in Appendix D. We used the Adam optimizer [20] with a learning rate of $0.002$, and used Cutout [10] and RandomAugment [8] for strong data augmentation, following [18]. Similar to [3, 4], we evaluated the performance of the proposed algorithm using an exponential moving average of the parameters over iterations with a decay rate of $0.999$, instead of scheduling the learning rate. In Tables 1-5, we used the overall accuracy and the accuracy only for minority classes as performance measures. We repeated the experiments five times under the main setting, and three times under the step imbalance and other settings of $\beta$ and $\gamma$. We report the average and standard deviation of the performance measures over repeated experiments. For the vanilla algorithm, FixMatch+DARP+cRT, and ReMixMatch+DARP+cRT, which suffered from overfitting, we measured performance every 500 iterations and recorded the best performance. Further details of the experimental setup are described in Appendix E.

## 4.2 Experimental results

The performance of the competing algorithms under the main setting are summarized in Table 1. We can observe that the proposed algorithm achieved the highest overall performance, with improved performance for minority classes. Interestingly, VAT, an SSL algorithm, showed similar performance to the vanilla algorithm, and worse performance than BALMS, a CIL algorithm. Similarly, FixMatch and ReMixMatch, which do not consider class imbalance, showed poor performance for minority classes. Although BALMS mitigated class imbalance, it produced poor overall performance, as it did not use unlabeled data for training. This demonstrates the importance of using unlabeled data for training, even in the class-imbalanced setting. FixMatch+CReST+PDA and ReMixMatch+CReST+PDA mitigated class imbalance by using unlabeled data points classified as the minority classes with a higher probability, but produced lower performance than the proposed algorithm. This may be because even if all unlabeled data points classified as minority classes are additionally used for training, their amount is still less than that of the data in majority classes, while the proposed algorithm uses class-balanced minibatches by generating the $0/1$ mask. Fixmatch+DARP and ReMix-Match+DARP slightly mitigated class imbalance by refining biased pseudo-labels, but resulted in lower performance than the proposed algorithm. This may be because even perfect pseudo labels cannot change the underlying class-imbalanced distribution of the training data. By additionally using a rebalancing technique cRT, FixMatch(ReMixMatch)+DARP+cRT performed better than FixMatch(ReMixMatch)+DARP. However, FixMatch(ReMixMatch)+DARP+cRT still performed worse than FixMatch(ReMixMatch)+ABC, although it also uses high-quality representations learned by FixMatch(ReMixMatch) and techniques for mitigating class imbalance. The superior performance of FixMatch(ReMixMatch)+ABC over FixMatch(ReMixMatch)+DARP+cRT is probably because FixMatch(ReMixMatch)+ABC was trained end-to-end, and the ABC was also trained using unlabeled data. We discuss this in more detail in Sections 4.4 and 4.5. Overall, the algorithms combined with

ReMixMatch performed better than the algorithms combined with FixMatch. In addition to the overall accuracy and minority-class-accuracy, we also compared the performance of the competing algorithms in terms of the geometric mean (G-mean) of class-wise accuracy under the main setting in Appendix F.

Table 1: Overall accuracy/minority-class-accuracy under the main setting

| | CIFAR-10-LT | SVHN-LT | CIFAR-100-LT |
|---|---|---|---|
| Algorithm | $\gamma = 100, \beta = 20\%$ | $\gamma = 100, \beta = 20\%$ | $\gamma = 20, \beta = 40\%$ |
| Vanilla | $55.3_{\pm1.30}$ / $33.9_{\pm1.88}$ | $77.0_{\pm0.67}$ / $63.3_{\pm1.25}$ | $40.1_{\pm1.15}$ / $25.2_{\pm0.95}$ |
| VAT [24] | $55.3_{\pm0.88}$ / $28.2_{\pm1.55}$ | $81.3_{\pm0.47}$ / $68.2_{\pm0.88}$ | $40.4_{\pm0.34}$ / $24.8_{\pm0.38}$ |
| BALMS [27] | $70.7_{\pm0.59}$ / $69.8_{\pm1.03}$ | $87.6_{\pm0.53}$ / $85.0_{\pm0.67}$ | $50.2_{\pm0.54}$ / $42.9_{\pm1.03}$ |
| FixMatch [29] | $72.3_{\pm0.33}$ / $53.8_{\pm0.63}$ | $88.0_{\pm0.30}$ / $79.4_{\pm0.54}$ | $51.0_{\pm0.20}$ / $32.8_{\pm0.41}$ |
| w/ CReST+PDA [34] | $76.6_{\pm0.46}$ / $61.4_{\pm0.85}$ | $89.1_{\pm0.69}$ / $81.7_{\pm1.18}$ | $51.6_{\pm0.29}$ / $36.4_{\pm0.46}$ |
| w/ DARP [18] | $73.7_{\pm0.98}$ / $57.0_{\pm2.12}$ | $88.6_{\pm0.19}$ / $80.5_{\pm0.54}$ | $51.4_{\pm0.37}$ / $33.9_{\pm0.77}$ |
| w/ DARP+cRT [18] | $78.1_{\pm0.89}$ / $66.6_{\pm1.55}$ | $89.9_{\pm0.44}$ / $83.5_{\pm0.61}$ | $54.7_{\pm0.46}$ / $41.2_{\pm0.42}$ |
| w/ ABC | $\mathbf{81.1}_{\pm0.82}$ / $\mathbf{72.0}_{\pm1.77}$ | $\mathbf{92.0}_{\pm0.38}$ / $\mathbf{87.9}_{\pm0.73}$ | $\mathbf{56.3}_{\pm0.19}$ / $\mathbf{43.4}_{\pm0.42}$ |
| ReMixMatch [3] | $73.7_{\pm0.39}$ / $55.9_{\pm0.87}$ | $89.8_{\pm0.42}$ / $82.8_{\pm0.68}$ | $54.0_{\pm0.29}$ / $37.1_{\pm0.37}$ |
| w/ CReST+PDA [34] | $75.7_{\pm0.34}$ / $59.6_{\pm0.76}$ | $90.9_{\pm0.20}$ / $85.2_{\pm0.39}$ | $54.6_{\pm0.48}$ / $38.1_{\pm0.69}$ |
| w/ DARP [18] | $74.4_{\pm0.41}$ / $56.9_{\pm0.67}$ | $90.2_{\pm0.22}$ / $83.5_{\pm0.40}$ | $54.5_{\pm0.33}$ / $37.7_{\pm0.58}$ |
| w/ DARP+cRT [18] | $78.5_{\pm0.61}$ / $66.4_{\pm1.68}$ | $92.1_{\pm0.48}$ / $87.6_{\pm0.75}$ | $55.1_{\pm0.45}$ / $43.6_{\pm0.58}$ |
| w/ ABC | $\mathbf{82.4}_{\pm0.45}$ / $\mathbf{75.7}_{\pm1.18}$ | $\mathbf{93.9}_{\pm0.16}$ / $\mathbf{92.5}_{\pm0.4}$ | $\mathbf{57.6}_{\pm0.26}$ / $\mathbf{46.7}_{\pm0.50}$ |

To evaluate the performance of the proposed algorithm in various settings, we conducted experiments using ReMixMatch, FixMatch, and the CISSL algorithms considered in Table 1, while changing the ratio of class imbalance $\gamma$ and the ratio of the amount of labeled data $\beta$. The results for CIFAR-10 are presented in Table 2, and the results for SVHN and CIFAR-100 are presented in Appendix G. In Table 2, we can observe that the proposed algorithm achieved the highest overall accuracy with greatly improved performance for minority classes for all settings. Because FixMatch+DARP+cRT and ReMixMatch+DARP+cRT do not use unlabeled data for classifier tuning, the difference in performance between FixMatch(ReMixMatch)+DARP+cRT and the proposed algorithm increased as the ratio of the amount of labeled data $\beta$ decreased and as the ratio of class imbalance $\gamma$ increased. In addition, the difference in performance between FixMatch(ReMixMatch)+CReST+PDA and the proposed algorithm tended to increase as the ratio of class imbalance $\gamma$ increased, because the difference between the number of labeled data points belonging to the majority classes and the number of unlabeled data points classified as the minority classes increases with $\gamma$.

Table 2: Overall accuracy/minority-class accuracy for CIFAR-10 under various settings

| | CIFAR-10-LT | | | |
|---|---|---|---|---|
| Algorithm | $\gamma = 100, \beta = 10\%$ | $\gamma = 100, \beta = 30\%$ | $\gamma = 50, \beta = 20\%$ | $\gamma = 150, \beta = 20\%$ |
| FixMatch [29] | $70.0_{\pm0.59}$ / $48.9_{\pm1.04}$ | $74.9_{\pm0.63}$ / $58.2_{\pm1.28}$ | $81.2_{\pm0.07}$ / $70.7_{\pm0.36}$ | $68.5_{\pm0.60}$ / $45.8_{\pm1.15}$ |
| w/ CReST+PDA [34] | $73.9_{\pm0.40}$ / $58.9_{\pm1.14}$ | $77.6_{\pm0.73}$ / $64.0_{\pm1.39}$ | $83.3_{\pm0.10}$ / $75.7_{\pm0.39}$ | $70.0_{\pm0.82}$ / $49.4_{\pm1.52}$ |
| w/ DARP+cRT [18] | $74.6_{\pm0.98}$ / $59.2_{\pm2.12}$ | $79.0_{\pm0.25}$ / $67.7_{\pm0.95}$ | $83.6_{\pm0.42}$ / $77.1_{\pm1.19}$ | $73.2_{\pm0.85}$ / $57.1_{\pm1.13}$ |
| w/ ABC | $\mathbf{77.2}_{\pm1.60}$ / $\mathbf{65.7}_{\pm2.85}$ | $\mathbf{81.5}_{\pm0.29}$ / $\mathbf{72.9}_{\pm0.96}$ | $\mathbf{85.2}_{\pm0.51}$ / $\mathbf{80.2}_{\pm0.64}$ | $\mathbf{77.1}_{\pm0.46}$ / $\mathbf{64.4}_{\pm0.92}$ |
| ReMixMatch [3] | $71.5_{\pm0.51}$ / $52.2_{\pm1.08}$ | $75.8_{\pm0.10}$ / $59.4_{\pm0.17}$ | $81.5_{\pm0.17}$ / $70.7_{\pm0.32}$ | $69.9_{\pm0.23}$ / $48.4_{\pm0.60}$ |
| w/ CReST+PDA [34] | $73.8_{\pm0.32}$ / $56.6_{\pm0.43}$ | $78.6_{\pm0.73}$ / $64.8_{\pm1.49}$ | $83.9_{\pm0.26}$ / $75.4_{\pm0.52}$ | $71.3_{\pm0.77}$ / $50.8_{\pm1.59}$ |
| w/ DARP+cRT [18] | $75.9_{\pm1.20}$ / $62.1_{\pm3.10}$ | $81.0_{\pm0.16}$ / $70.7_{\pm0.72}$ | $84.5_{\pm0.80}$ / $77.8_{\pm1.67}$ | $73.9_{\pm0.59}$ / $57.4_{\pm1.45}$ |
| w/ ABC | $\mathbf{79.8}_{\pm0.36}$ / $\mathbf{70.8}_{\pm0.92}$ | $\mathbf{84.3}_{\pm1.03}$ / $\mathbf{80.6}_{\pm0.97}$ | $\mathbf{87.5}_{\pm0.31}$ / $\mathbf{84.6}_{\pm1.19}$ | $\mathbf{80.6}_{\pm0.66}$ / $\mathbf{72.1}_{\pm1.51}$ |

We also conducted experiments under a step-imbalance setting, where the class imbalance was more noticeable. This setting assumes a more severely imbalanced class distribution than the LT imbalance settings, because half of the classes have very scarce data. The experimental results for CIFAR-10 are presented in Table 3, and the results for SVHN and CIFAR-100 are presented in Appendix H. In Table 3, we can see that the proposed algorithm achieved the best performance, and the performance margin is greater than that of the LT imbalance settings. ReMixMatch+CReST+PDA showed relatively low performance compared to the other algorithms.

Table 3: Overall accuracy/minority-class accuracy on CIFAR-10 under a step imbalance setting

| CIFAR-10-Step, $\gamma = 100$, $\beta = 20\%$ | | | | |
|---|---|---|---|---|
| Algorithm | w/ - | w/ CReST+PDA [34] | w/ DARP+cRT [18] | w/ ABC |
| FixMatch [29] | $54.0_{\pm 0.84}$ / $11.8_{\pm 1.71}$ | $71.1_{\pm 0.78}$ / $48.2_{\pm 2.26}$ | $69.8_{\pm 1.51}$ / $45.1_{\pm 2.70}$ | $\mathbf{75.9}_{\pm 0.49}$ / $\mathbf{57.0}_{\pm 1.07}$ |
| ReMixMatch [3] | $60.8_{\pm 0.10}$ / $25.1_{\pm 1.28}$ | $64.6_{\pm 0.97}$ / $33.5_{\pm 2.05}$ | $72.3_{\pm 1.77}$ / $50.6_{\pm 3.53}$ | $\mathbf{76.4}_{\pm 1.70}$ / $\mathbf{65.7}_{\pm 1.30}$ |

To evaluate the performance of the proposed algorithm on a large-scale dataset, we also conducted experiments on the LSUN dataset [37], which is naturally a long-tailed dataset. Among the algorithms considered in Tables 2 and 3, those combined with CReST were excluded for comparison, because CReST requires loading of the whole unlabeled data in the repeated process of updating pseudo-labels, which is not possible for the large-scale LSUN dataset. Instead, we additionally considered FixMatch+cRT and ReMixMatch+cRT for comparison. The experimental results are presented in Table 4. The proposed algorithm showed better performance than the other baseline algorithms. DARP resulted in degradation of the performance, possibly because the scale of the LSUN dataset is very large. Specifically, DARP solves a convex optimization with all unlabeled data points to refine the pseudo labels. As the scale of the unlabeled dataset increases, this optimization problem becomes more difficult to solve and, consequently, the pseudo-labels could be refined inaccurately. Unlike the results for other datasets, the algorithms combined with FixMatch performed better than the algorithms combined with ReMixMatch.

Table 4: Overall accuracy/minority-class accuracy for the large-scale LSUN dataset

| LSUN, $\gamma = 100$, $\beta = 20\%$ | | | | | |
|---|---|---|---|---|---|
| Algorithm | w/ - | w/ cRT [17] | w/ DARP [18] | w/ DARP+cRT [18] | w/ ABC |
| FixMatch [29] | 73.1 / 55.3 | 77.0 / 71.5 | 71.0 / 51.8 | 75.8 / 69.5 | **78.9 / 75.5** |
| ReMixMatch [3] | 69.4 / 49.1 | 75.4 / 69.5 | 65.6 / 44.1 | 72.1 / 67.5 | **76.9 / 69.5** |

### 4.3 Complexity of the proposed algorithm

The proposed algorithm requires additional parameters for the ABC, but the number of the additional parameters is negligible compared to the number of parameters of the backbone. For example, the ABC additionally required only $0.09\%$ and $0.87\%$ of the number of backbone parameters for CIFAR-10 with 10 classes and CIFAR-100 with 100 classes, respectively. Moreover, because the ABC shares the representation layer of the backbone, it does not significantly increase the memory usage and training time. Furthermore, we could train the proposed algorithm on the large-scale LSUN dataset without a significant increase in computation cost, because the entire training procedure could be carried out using minibatches of data. In contrast, the algorithms combined with DARP required convex optimization for all pseudo-labels, which significantly increased the computation cost as the number of classes or the amount of data increased. Similarly, it required significant time to train the algorithms combined with CReST, because CReST requires iterative re-training with a labeled set expanded by adding unlabeled data points with pseudo-labels. We present the floating point operations per second (FLOPS) for each algorithm using Nvidia Tesla-V100 in Appendix I.

### 4.4 Qualitative analysis of high-quality representations and balanced classification

The ABC can use high-quality representations learned by the backbone when performing balanced classification. To verify this, in Figure 3, we present t-distributed stochastic neighbor embedding (t-SNE) [31] of the representations of the CIFAR-10 test set learned by the ABC (without SSL backbone), FixMatch+ABC, and ReMixMatch+ABC on CIFAR-10-LT under the main setting. Different colors indicate different classes. As expected, "ABC (without SSL backbone)" failed to learn class-separable representations because sufficient data were not used for training while using the $0/1$ mask. In contrast, by training the backbone (FixMatch or ReMixMatch) together with the ABC, the proposed algorithm could use the entire data and learn high-quality representations. In this example, ReMixMatch produced more separable representations than FixMatch, which shows that the choice of the backbone affects the performance of the proposed algorithm, as expected.

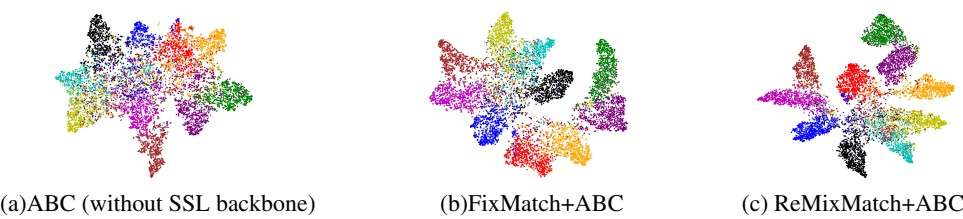

| (a)ABC (without SSL backbone) | (b)FixMatch+ABC | (c) ReMixMatch+ABC |

Figure 3: t-SNE of the proposed algorithm and the ABC (without SSL backbone)

The proposed algorithm can also mitigate class imbalance by using the ABC. To verify this, we compare the confusion matrices of the predictions on the test set of CIFAR-10 using ReMixMatch, ReMixMatch+DARP+cRT, and ReMiMatch+ABC trained on CIFAR-10 under the main setting in Figure 4. In the confusion matrices, the value in the $i$th row and the $j$th column represents the ratio of the amount of data belonging to the $i$th class to the amount of data predicted as the $j$th class. Each cell has a darker red color when the ratio is larger. We can see that ReMixMatch often misclassified data points in the minority classes (e.g., classes 8 and 9 into classes 0 and 1). This may be because ReMixMatch does not consider class imbalance, and thus biased pseudo-labels were used for training. ReMixMatch+DARP+cRT produced a more balanced class-distribution compared to ReMixMatch by additionally using DARP+cRT. However, a significant number of data points in the minority classes were still misclassified as majority classes. In contrast, ReMixMatch+ABC classified the test data points in the minority classes with higher accuracy, and produced a significantly more balanced class distribution than ReMixMatch+DARP+cRT, as shown in Figure 4 (c). As both ReMixMatch+DARP+cRT and ReMixMatch+ABC use ReMixMatch to learn representations, the performance gap between these two algorithms results from the different characteristics of the ABC versus DARP+cRT as follows. First, DARP+cRT does not use unlabeled data for training its classifier after representations learning is completed, whereas the ABC uses unlabeled data with unbiased pseudo-labels for its training. Second, whereas DARP+cRT decouples the learning of representations and training of a classifier, the ABC is trained end-to-end interactively with representations learned by the backbone. We also present the confusion matrices of the predictions on the test set of CIFAR-10 using FixMatch, FixMatch+DARP+cRT, and FixMatch+ABC as well as the confusion matrices of the pseudo-labels on the same dataset using ReMixMatch, ReMixMatch+DARP+cRT, ReMix-Match+ABC, FixMatch, FixMatch+DARP+cRT, and FixMatch+ABC in Appendix J. Moreover, we compare the ABC and the classifier of DARP+cRT in more detail using the validation loss plots in Appendix K.

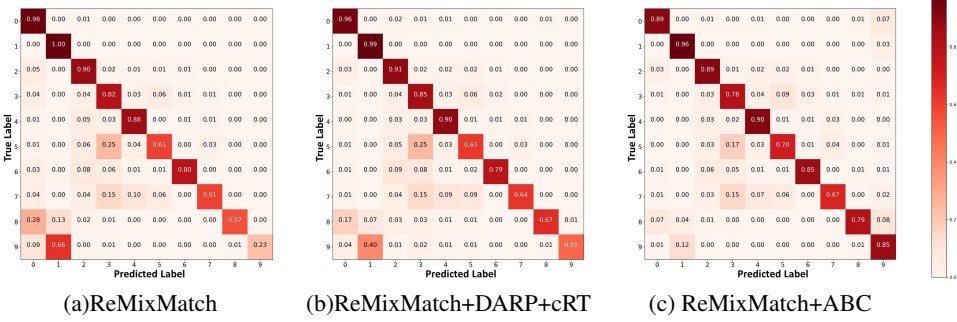

| (a)ReMixMatch | (b)ReMixMatch+DARP+cRT | (c) ReMixMatch+ABC |

Figure 4: Confusion matrices of the predictions on the test set of CIFAR-10

## 4.5 Ablation study

We conducted an ablation study on CIFAR-10-LT in the main setting to investigate the effect of each element of the proposed algorithm. The results for ReMixMatch+ABC are presented in Table 5, where each row indicates the proposed algorithm with the described conditions in that row. The results are summarized as follows. 1) If we did not gradually decrease the parameter of the Bernoulli distribution $\mathcal{B}\left(\cdot\right)$ when conducting consistency regularization, then an overbalance problem occurred

because of unlabeled data misclassified as minority classes. 2) Without consistency regularization for the ABC, the decision boundary did not clearly separate each class. 3) Without using the 0/1 mask for $L_{cls}$ and $L_{con}$, the ABC was trained to be biased toward the majority classes. 4) Without confidence threshold $\tau$ for consistency regularization, training became unstable and, consequently, the ABC was trained to be biased toward certain classes. 5) Similarly, if hard pseudo-labels, instead of soft pseudo-labels, were used for consistency regularization, then the ABC was biased toward certain classes. 6) If the ABC was solely used without the backbone, the performance decreased because the ABC could not use high-quality representations learned by the backbone. 7) When we used a re-weighting technique [13] instead of a mask for the ABC, training became unstable because of abnormally large gradients calculated for training on the data of the minority classes. 8) The decoupled training of the backbone and ABC resulted in decreased classification performance, as was also analyzed in Section 4.4. Similarly, we present the results of the ablation study for FixMatch+ABC in Appendix L.

Table 5: Ablation study for ReMixMatch+ABC on CIFAR-10-LT, $\gamma = 100$, $\beta = 20\%$

| Ablation study | Overall | Minority |
|---|---|---|
| ReMixMatch+ABC (proposed algorithm) | **82.4** | **75.7** |
| Without gradually decreasing the parameter of $\mathcal{B}(\cdot)$ for consistency regularization | 81.8 | 74.6 |
| Without consistency regularization for the ABC | 79.4 | 66.9 |
| Without using the 0/1 mask for the consistency regularization loss $L_{con}$ | 79.0 | 69.2 |
| Without using the 0/1 mask for the classification loss $L_{cls}$ | 74.4 | 57.8 |
| Without using the confidence threshold $\tau$ for consistency regularization | 74.3 | 75.4 |
| Using hard pseudo labels for consistency regularization | 70.2 | 75.1 |
| Without training backbone (ABC without SSL backbone) | 68.7 | 56.2 |
| Training the ABC with a re-weighting technique | 81.2 | 74.1 |
| Decoupled training of the backbone and ABC | 79.5 | 72.3 |

## 5   Conclusion

We introduced the ABC, which is attached to a state-of-the-art SSL algorithm, for CISSL. The ABC can utilize high-quality representations learned by the backbone, while being trained to make class-balanced predictions. The ABC also utilizes unlabeled data by conducting consistency regularization in a modified way for class-imbalance problems. The experimental results obtained under various settings demonstrate that the proposed algorithm outperforms the baseline algorithms. We also conducted a qualitative analysis and an ablation study to verify the contribution of each element of the proposed algorithm. The proposed algorithm assumes that the labeled and unlabeled data are class-imbalanced to the same extent. In the future, we plan to release this assumption by adopting a module for estimating class distribution. Deep learning algorithms can be applied to many societal problems. However, if the training data are imbalanced, the algorithms could be trained to make socially biased decisions in favor of the majority groups. The proposed algorithm can contribute to solving these issues. However, there is also a potential risk that the proposed algorithm could be used as a tool to identify minorities and discriminate against them. It should be ensured that the proposed method cannot be used for any purpose that may have negative social impacts.

## Acknowledgments

This work was supported by the National Research Foundation of Korea (NRF) grant funded by the Korea government (MSIT) (2018R1C1B6004511, 2020R1A4A10187747).

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
