# Supplementary Material for the Paper entitled "ABC: Auxiliary Balanced Classifier for Class-Imbalanced Semi-Supervised Learning"

**Hyuck Lee   Seungjae Shin   Heeyoung Kim**
Department of Industrial and Systems Engineering, KAIST
{dlgur0921, tmdwo0910, heeyoungkim}@kaist.ac.kr

## A   Overall procedure of consistency regularization for ABC

Figure 1 illustrates the overall procedure of consistency regularization for the ABC. Detailed procedure is described in Section 3.4 of the main paper.

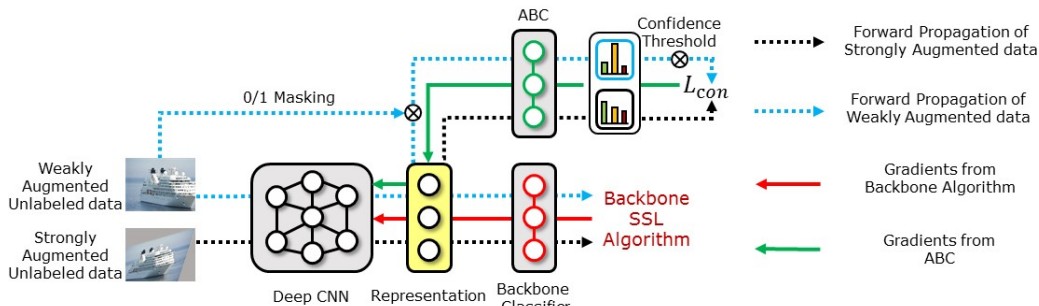

Figure 1: Overall procedure of consistency regularization for the ABC

## B   Pseudo code of the proposed algorithm

The pseudo code of the proposed algorithm is presented in Algorithm 1. The for loop (lines 2 14) can be run in parallel. The classification loss $L_{cls}$ and consistency regularization loss $L_{con}$ are expressed in detail in Sections 3.3 and 3.4 of the main paper.

## C   Two types of class imbalance for the considered datasets

Two types of class imbalance for the considered datasets are illustrated in Figure 2. For both types of imbalance, we set $\gamma = 100$, $N_1 = 1000$, and $\beta = 20\%$. In Figure 2 (b), we can see that each minority class has a very small amount of data. Existing SSL algorithms can be significantly biased toward majority classes under step imbalanced settings.

35th Conference on Neural Information Processing Systems (NeurIPS 2021), Sydney, Australia.

**Algorithm 1** Pseudo code of the proposed algorithm

---

**Input:** $\mathcal{MB}_{\mathcal{X}} = \{(x_b, y_b) : b \in (1, ..., B)\} \subset \mathcal{X}, \mathcal{MB}_{\mathcal{U}} = \{(u_b) : b \in (1, ..., B)\} \subset \mathcal{U}$
**Output:** Classification model $f : \mathbb{R}^d \rightarrow \{1, ...L\}$
**Parameters :** $\theta$ (Parameters of Wide ResNet-28-2 and ABC)

1: **while** Training **do**
2:     **for** $b = 1$ to $B$ **do**
3:         $\alpha(x_b) = \text{Augment}(x_b)$
4:         $\alpha(u_b) = \text{WeakAugment}(u_b)$
5:         $\mathcal{A}_k(u_b) = \text{StrongAugment}_k(x_b), k = 1, 2$
6:         Predicted class distribution for $\alpha(x_b) = p_s(y|\alpha(x_b))$
7:         Generate 0/1 mask $M(x_b)$.
8:         Soft pseudo label $q_b = p_s(y|\alpha(u_b))$
9:         **if** $\max(q_b) \geq 0.95$ **then**
10:             Predicted class distribution for $\mathcal{A}_k(u_b) = p_s(y|\mathcal{A}_k(u_b)), k = 1, 2$
11:             Generate 0/1 mask $M(u_b)$.
12:         **end if**
13:         Loss from the backbone $L_{back} \mathrel{+}= \text{backbone}(\alpha(x_b), \alpha(u_b), \mathcal{A}_k(u_b))$
14:     **end for**
15:     Calculate the classification loss $L_{cls}$ and consistency regularization loss $L_{con}$.
16:     Total Loss $L_{total} = L_{cls} + L_{con} + L_{back}$
17:     $\Delta\theta \propto \nabla_\theta L_{total}, \quad \theta \leftarrow \theta + \Delta\theta$
18: **end while**

---

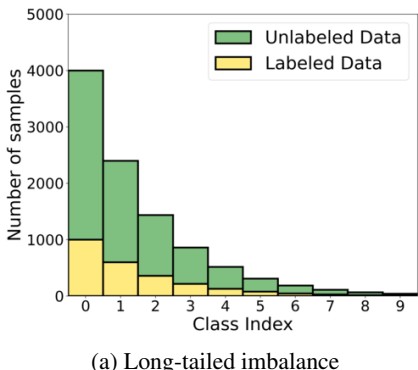

(a) Long-tailed imbalance

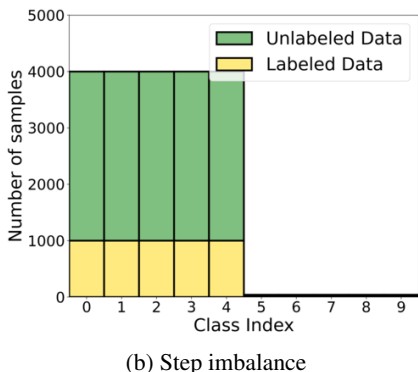

(b) Step imbalance

Figure 2: Long-tailed imbalance and step imbalance

## D   Specification of the confidence threshold $\tau$

Table 1: Mean and standard deviation (STD) of validation accuracy during the last 50 epochs

| ReMixMatch+ABC | CIFAR-10-LT,   $\gamma = 100$,   $\beta = 20\%$ | | | | | | |
|---|---|---|---|---|---|---|---|
| $\tau$ | 1 | 0.98 | **0.95** | 0.9 | 0.85 | 0.8 | 0.75 | 0.7 |
| Mean and STD | 78.9, 0.36 | 81.8, 0.34 | **82.3, 0.2** | 81.3, 0.32 | 81.5, 0.39 | 81.2, 0.63 | 80.0, 2.87 | 79.0, 5.76 |

In general, the confidence threshold $\tau$ should be set high enough, but not too high. If $\tau$ is low, training becomes unstable because many misclassified unlabeled data points would be used for training. However, if $\tau$ is too high, most of the unlabeled data points would not be used for consistency regularization. Based on these insights, we set $\tau$ as 0.95 in our experiments. We confirmed via experiments that this value of $\tau$ enabled high accuracy as well as stability. Specifically, we conducted experiments on CIFAR-10-LT for the main setting while changing the value of $\tau$. We measured the validation accuracy of ReMixMatch+ABC during the last 50 epochs (1 epoch=500 iterations) of training and calculated the mean and standard deviation (STD) of these values. As can be seen from Table 1, the proposed algorithm achieved the highest mean and lowest STD of the validation accuracy

when $\tau$ was 0.95. When $\tau$ was set higher or lower than 0.95, the mean of the validation accuracy decreased. In particular, as the value of $\tau$ decreased from 0.95, the STD increased rapidly, indicating instability of the training.

# E    Further details of the experimental setup

We describe further details of the experimental setup. To train the ReMixMatch, we gradually increased the coefficient of the loss associated with the unlabeled data points, following [4]. We found that without this gradual increase, the validation loss of the ReMixMatch did not converge. To train the FixMatch, we used the labeled dataset once more as an unlabeled dataset by removing the labels for the experiments using CIFAR-100 following the previous study [5], but not for the experiments using CIFAR-10 and SVHN, because it did not improve the performance. We followed the default settings for the ReMixMatch [1] and FixMatch [5], unless mentioned otherwise.

To train the ABC, we also gradually decreased the parameter of $\mathcal{B}(\cdot)$ for calculating the classification loss in the experiments using CIFAR-10 and SVHN under the step imbalanced setting. This prevents unstable training by allowing each labeled data point of the majority classes to be more frequently used for training.

# F    Geometric mean (G-mean) of class-wise accuracy for the main setting

Table 2: Performance comparison using G-mean for the main setting

|  | CIFAR-10-LT | SVHN-LT | CIFAR-100-LT |
|---|---|---|---|
| Algorithm | $\gamma = 100, \beta = 20\%$ | $\gamma = 100, \beta = 20\%$ | $\gamma = 20, \beta = 40\%$ |
| FixMatch [5] | 62.0 | 87.3 | 38.5 |
| w/ CReST+PDA [6] | 74.4 | 88.6 | 42.3 |
| w/ DARP [4] | 71.5 | 87.6 | 40.4 |
| w/ DARP+cRT [4] | 76.7 | 89.8 | 47.0 |
| w/ ABC | **80.5** | **91.8** | **49.0** |
| ReMixMatch [1] | 62.5 | 89.5 | 41.2 |
| w/ CReST+PDA [6] | 72.2 | 90.7 | 43.1 |
| w/ DARP [4] | 71.9 | 89.7 | 42.5 |
| w/ DARP+cRT [4] | 77.9 | 92.0 | 48.3 |
| w/ ABC | **81.9** | **93.8** | **50.8** |

To evaluate whether the proposed algorithm performs in a balanced way for all classes, we also measured the performance for the main setting using the geometric mean (G-mean) of class-wise accuracy with correction to avoid zeroing. We set the hyperparameter for the correction to avoid zeroing as 1%, which indicates that the minimum class-wise accuracy is 1%. The results in Table 2 demonstrates that the proposed algorithm performed in a balanced way.

# G    Experimental results on SVHN and CIFAR-100 for various settings

For the experiments using SVHN with $\gamma = 150$ and $\beta = 20\%$, the solution of the convex optimization problem of ReMixMatch+DARP+cRT for refining the pseudo labels did not converge, and thus we could not measure the performance. The experimental results for SVHN and CIFAR-100 under various settings showed the same trend as those for CIFAR-10, which is described in Section 4.2 of the main paper.

Table 3: Overall accuracy/minority-class-accuracy on SVHN for various settings

| | SVHN-LT | | | |
|---|---|---|---|---|
| Algorithm | $\gamma = 100, \beta = 10\%$ | $\gamma = 100, \beta = 30\%$ | $\gamma = 50, \beta = 20\%$ | $\gamma = 150, \beta = 20\%$ |
| FixMatch [5] | $88.5_{\pm 0.25}/80.3_{\pm 0.42}$ | $88.7_{\pm 0.36}/80.7_{\pm 0.65}$ | $91.1_{\pm 0.18}/85.3_{\pm 0.28}$ | $85.6_{\pm 0.17}/74.6_{\pm 0.43}$ |
| w/ CReST + PDA [6] | $89.2_{\pm 0.43}/81.7_{\pm 0.95}$ | $89.9_{\pm 0.36}/83.0_{\pm 0.37}$ | $91.7_{\pm 0.86}/87.6_{\pm 0.53}$ | $86.7_{\pm 0.89}/76.7_{\pm 1.70}$ |
| w/ DARP + cRT [4] | $89.3_{\pm 0.33}/83.9_{\pm 0.47}$ | $90.7_{\pm 0.28}/84.8_{\pm 0.37}$ | $92.1_{\pm 0.30}/87.7_{\pm 0.44}$ | $88.0_{\pm 0.74}/80.1_{\pm 1.88}$ |
| w/ ABC | $\mathbf{92.3}_{\pm 0.38}/\mathbf{88.7}_{\pm 0.92}$ | $\mathbf{92.3}_{\pm 0.34}/\mathbf{88.3}_{\pm 0.49}$ | $\mathbf{93.5}_{\pm 0.17}/\mathbf{90.7}_{\pm 0.25}$ | $\mathbf{91.2}_{\pm 0.15}/\mathbf{86.2}_{\pm 0.15}$ |
| ReMixMatch [1] | $89.2_{\pm 0.17}/81.7_{\pm 0.41}$ | $90.7_{\pm 0.15}/84.5_{\pm 0.46}$ | $92.4_{\pm 0.21}/87.8_{\pm 0.48}$ | $88.6_{\pm 0.16}/80.4_{\pm 0.42}$ |
| w/ CReST + PDA [6] | $89.8_{\pm 0.12}/83.0_{\pm 0.08}$ | $91.2_{\pm 0.17}/85.3_{\pm 0.24}$ | $93.3_{\pm 0.02}/90.0_{\pm 0.36}$ | $88.8_{\pm 0.41}/80.7_{\pm 0.82}$ |
| w/ DARP + cRT [4] | $91.7_{\pm 0.26}/86.6_{\pm 0.45}$ | $93.2_{\pm 0.08}/89.3_{\pm 0.21}$ | $93.6_{\pm 0.41}/90.4_{\pm 0.52}$ | $-/-$ |
| w/ ABC | $\mathbf{93.2}_{\pm 0.64}/\mathbf{92.2}_{\pm 0.44}$ | $\mathbf{94.4}_{\pm 0.37}/\mathbf{93.3}_{\pm 0.32}$ | $\mathbf{94.7}_{\pm 0.35}/\mathbf{93.5}_{\pm 0.56}$ | $\mathbf{93.2}_{\pm 0.46}/\mathbf{91.8}_{\pm 0.79}$ |

Table 4: Overall accuracy/minority-class-accuracy on CIFAR-100 for various settings

| | CIFAR-100-LT | | | |
|---|---|---|---|---|
| Algorithm | $\gamma = 20, \beta = 20\%$ | $\gamma = 20, \beta = 50\%$ | $\gamma = 10, \beta = 40\%$ | $\gamma = 30, \beta = 40\%$ |
| FixMatch [5] | $46.1_{\pm 0.23}/26.6_{\pm 0.34}$ | $52.3_{\pm 0.54}/34.7_{\pm 0.80}$ | $57.4_{\pm 0.15}/44.8_{\pm 0.17}$ | $47.6_{\pm 0.09}/27.6_{\pm 0.21}$ |
| w/ CReST + PDA [6] | $46.7_{\pm 0.49}/29.3_{\pm 0.54}$ | $52.7_{\pm 0.06}/37.4_{\pm 0.37}$ | $57.3_{\pm 0.23}/47.5_{\pm 0.22}$ | $48.5_{\pm 0.06}/30.0_{\pm 0.04}$ |
| w/ DARP + cRT [4] | $48.9_{\pm 0.11}/33.5_{\pm 0.17}$ | $55.9_{\pm 0.43}/43.5_{\pm 1.28}$ | $59.0_{\pm 0.40}/50.4_{\pm 1.09}$ | $51.3_{\pm 0.29}/36.4_{\pm 0.50}$ |
| w/ ABC | $\mathbf{49.7}_{\pm 0.40}/\mathbf{34.6}_{\pm 0.76}$ | $\mathbf{58.3}_{\pm 0.74}/\mathbf{46.7}_{\pm 1.12}$ | $\mathbf{61.6}_{\pm 0.15}/\mathbf{53.0}_{\pm 0.26}$ | $\mathbf{53.6}_{\pm 0.35}/\mathbf{38.8}_{\pm 0.69}$ |
| ReMixMatch [1] | $49.0_{\pm 0.29}/29.9_{\pm 0.42}$ | $54.4_{\pm 0.13}/37.8_{\pm 0.12}$ | $59.5_{\pm 0.20}/47.1_{\pm 0.42}$ | $51.0_{\pm 0.11}/32.0_{\pm 0.50}$ |
| w/ CReST + PDA [6] | $49.4_{\pm 0.32}/31.8_{\pm 0.15}$ | $54.4_{\pm 0.21}/38.6_{\pm 0.35}$ | $58.8_{\pm 0.08}/47.6_{\pm 0.24}$ | $51.9_{\pm 0.34}/33.5_{\pm 0.69}$ |
| w/ DARP + cRT [4] | $50.2_{\pm 0.40}/35.2_{\pm 0.55}$ | $54.6_{\pm 1.75}/44.8_{\pm 2.09}$ | $59.4_{\pm 1.04}/52.1_{\pm 0.71}$ | $52.8_{\pm 0.24}/38.4_{\pm 0.30}$ |
| w/ ABC | $\mathbf{52.5}_{\pm 0.10}/\mathbf{38.5}_{\pm 0.42}$ | $\mathbf{59.3}_{\pm 0.66}/\mathbf{49.5}_{\pm 1.02}$ | $\mathbf{63.5}_{\pm 0.29}/\mathbf{57.1}_{\pm 0.06}$ | $\mathbf{55.4}_{\pm 0.46}/\mathbf{42.8}_{\pm 0.67}$ |

## H  Experimental results on SVHN and CIFAR-100 for the step imbalanced setting

Experimental results for SVHN and CIFAR-100 under the step imbalanced setting showed the same tendency as that for CIFAR-10, which is described in Section 4.2 of the main paper.

Table 5: Overall accuracy/minority-class-accuracy on SVHN for the step imbalanced setting

| | SVHN-Step, $\quad \gamma = 100, \quad \beta = 20\%$ | | | |
|---|---|---|---|---|
| Algorithm | w/ - | w/ CReST + PDA [6] | w/ DARP + cRT [4] | w/ ABC |
| FixMatch [5] | $79.8_{\pm 1.34}/61.5_{\pm 2.76}$ | $86.6_{\pm 0.19}/76.3_{\pm 0.23}$ | $85.9_{\pm 0.28}/74.3_{\pm 0.37}$ | $\mathbf{91.2}_{\pm 0.15}/\mathbf{85.6}_{\pm 0.35}$ |
| ReMixMatch [1] | $82.7_{\pm 0.42}/67.4_{\pm 0.81}$ | $85.9_{\pm 0.13}/73.9_{\pm 0.16}$ | $90.5_{\pm 1.13}/84.3_{\pm 1.86}$ | $\mathbf{91.3}_{\pm 1.61}/\mathbf{89.8}_{\pm 0.95}$ |

Table 6: Overall accuracy/minority-class-accuracy on CIFAR-100 for the step imbalanced setting

| | CIFAR-100-Step, $\quad \gamma = 20, \quad \beta = 40\%$ | | | |
|---|---|---|---|---|
| Algorithm | w/ - | w/ CReST + PDA [6] | w/ DARP + cRT [4] | w/ ABC |
| FixMatch [5] | $46.7_{\pm 0.29}/15.0_{\pm 0.26}$ | $49.9_{\pm 0.24}/26.7_{\pm 0.41}$ | $50.7_{\pm 0.61}/28.8_{\pm 2.04}$ | $\mathbf{54.7}_{\pm 0.06}/\mathbf{32.1}_{\pm 0.12}$ |
| ReMixMatch [1] | $47.3_{\pm 0.12}/16.5_{\pm 1.06}$ | $48.5_{\pm 0.18}/19.2_{\pm 0.35}$ | $53.6_{\pm 0.28}/35.0_{\pm 1.04}$ | $\mathbf{56.0}_{\pm 0.46}/\mathbf{38.3}_{\pm 0.55}$ |

## I  Floating point operations per second (FLOPS) of each algorithm

As we mentioned in Section 4.3 of the main paper, computation cost required for the algorithms combined with DARP increased as the number of classes or the amount of data increased. In contrast, computation cost required for the proposed algorithm did not significantly increased because the whole training procedure can be carried out using minibatches. FLOPS of FixMatch+CReST and ReMixMatch+CReST are the same as those of FixMatch and ReMixMatch, but the algorithms combined with CReST required iterative re-training with a labeled set expanded by adding unlabeled

data points with pseudo labels. We measured FLOPS using Nvidia Tesla-V100. For the experiments on CIFAR-10 and CIFAR-100, we used only one GPU, whereas we used four GPUs in parallel for the experiments on LSUN.

Table 7: FLOPS of each algorithm

| Algorithm | CIFAR-10 | CIFAR-100 | LSUN |
|---|---|---|---|
| FixMatch [5] | 14.5 iter/sec | 14.7 iter/sec | 2.6 iter/sec |
| FixMatch+DARP [4] | 12.0 iter/sec | 6.3 iter/sec | 0.4 iter/sec |
| FixMatch+ABC | 11.2 iter/sec | 11.0 iter/sec | 2.0 iter/sec |
| ReMixMatch [1] | 6.9 iter/sec | 6.9 iter/sec | 1.3 iter/sec |
| ReMixMatch+DARP [4] | 6.3 iter/sec | 4.5 iter/sec | 0.3 iter/sec |
| ReMixMatch+ABC | 5.8 iter/sec | 5.6 iter/sec | 0.9 iter/sec |

## J   Further qualitative analysis and quantitative comparison

Figure 3 (b) presents biased predictions of FixMatch [5], a recent SSL algorithm, trained on CIFAR-10 with the amount of Class 0 being 100 times more than that of Class 9 as depicted in Figure 3 (a). In contrast, Figure 3 (c) presents that the class distribution of the predicted labels became more balanced using the FixMatch+ABC trained on the same dataset. These results are consistent with those in Figure 1 of the main paper.

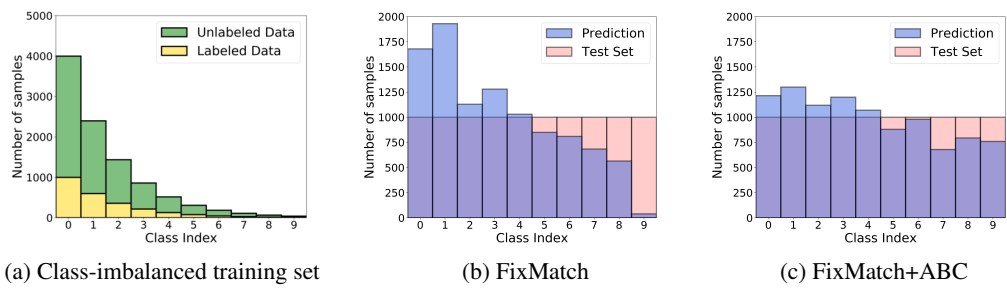

(a) Class-imbalanced training set      (b) FixMatch      (c) FixMatch+ABC

Figure 3: Predictions on a class-balanced test set of CIFAR-10 using FixMatch (b) and the FixMatch+ABC (c) trained on a class-imbalanced training set (a).

Because the use of the $0/1$ mask for the ABC plays a similar role of re-sampling techniques, we compare the representations of proposed algorithm with those of SMOTE (oversampling technique) [2]+CNN, and random undersampling [3]+CNN. Figure 4 (a), (b) and (c) present the t-SNE representations obtained using SMOTE+CNN, undersampling+CNN, and ABC only. Because re-sampling techniques can only be applied to labeled data, they cannot be combined with the SSL algorithms, and thus they were combined with CNN instead. SMOTE+CNN and undersampling+CNN learned less separable representations than the ABC only. These results show that using the $0/1$ mask instead of re-sampling techniques is more effective because we could utilize unlabeled data. In addition, the 0/1 mask enabled the ABC to be combined with the backbone, so that the ABC could use the high-quality representations learned by the backbone as shown in Figure 4 (d).

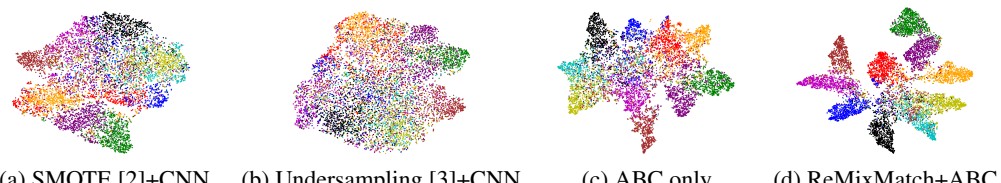

(a) SMOTE [2]+CNN    (b) Undersampling [3]+CNN    (c) ABC only    (d) ReMixMatch+ABC

Figure 4: t-SNE of the representations of the CIFAR-10 test set using re-sampling+CNN, ABC only, and ReMixMatch+ABC trained on CIFAR-10-LT, $\gamma = 100$, $\beta = 20\%$

We also compared the performance of the proposed algorithm with those of SMOTE+CNN and undersampling+CNN. The results in Table 8 show the importance of using unlabeled data for training and using the high-quality representations obtained from backbone.

Table 8: Performance of each algorithm in Figure 4 and FixMatch+ABC. The algorithms were trained on CIFAR-10-LT, $\gamma = 100$, $\beta = 20\%$ and tested on the test set of CIFAR-10.

| Performance of each algorithm in Figure 4 and FixMatch+ABC | Overall | Minority |
|---|---|---|
| ReMixMatch+ABC | **82.4** | **75.7** |
| FixMatch+ABC | 81.1 | 72.0 |
| Without training backbone (ABC only) | 68.7 | 56.2 |
| SMOTE+CNN | 60.8 | 46.1 |
| Undersampling+CNN | 48.7 | 55.8 |

Figure 5 presents the confusion matrices of FixMatch, FixMatch+DARP+cRT, and FixMatch+ABC trained on CIFAR-10-LT, $\gamma = 100$, $\beta = 20\%$. Similar to Figure 4 of the main paper, FixMatch and FixMatch+DARP+cRT often misclassified test data points in the minority classes (e.g., classes 8 and 9 into classes 0 and 1). In contrast, FixMatch+ABC classified the test data points in the minority classes with higher accuracy, and produced a significantly more balanced class-distribution than FixMatch and FixMatch+DARP+cRT.

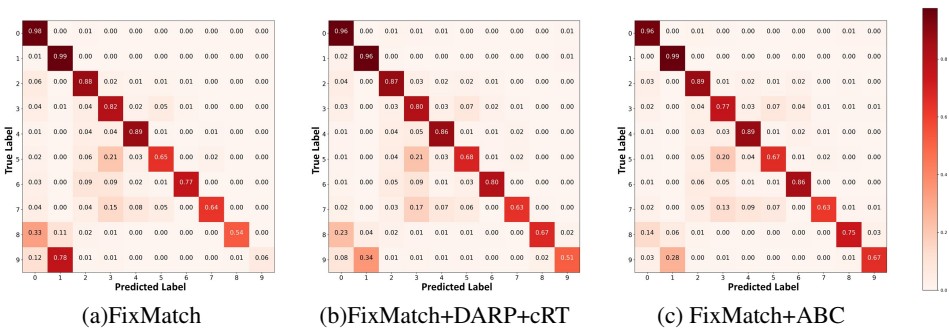

(a)FixMatch      (b)FixMatch+DARP+cRT      (c) FixMatch+ABC

Figure 5: Confusion matrices of the predictions on the test set of CIFAR-10

Figure 6 presents the confusion matrices of the predictions on the unlabeled data. Similar to Figure 5 and Figure 4 of the main paper, FixMatch+ABC and ReMixMatch+ABC classified the unlabeled data points in the minority classes with higher accuracy, and produced a significantly more balanced pseudo labels than other algorithms. By using these balanced pseudo labels for training, the proposed algorithm could make a more balanced prediction on the test set.

## K    Detailed comparison between the end-to-end training of the proposed algorithm and decoupled learning of representations and a classifier

Although FixMatch+DARP+cRT and ReMixMatch+DARP+cRT also use the representations learned by ReMixMatch [1] and FixMatch [5], they showed worse performance than the proposed algorithm. The performance gap between FixMatch(ReMixMatch)+DARP+cRT and the proposed algorithm results from the different characteristics of the ABC versus DARP+cRT as follows. First, whereas DARP+cRT decouples learning of representations and training of a classifier, the ABC is trained end-to-end interactively with representations that the backbone learns. Second, DARP+cRT does not use unlabeled data for training of its classifier after representations learning is finished, while the ABC is trained with unlabeled data to conduct consistency regularization so that decision boundaries can be placed in a low density region. To verify these reasons, we compare the validation loss graphs of the algorithms based on end-to-end training and decoupled learning of representations and a classifier in Figure 7. We recorded the validation loss of 100 epochs after the representations were fixed, where 1 epoch was set as 500 iterations. For the proposed algorithm, we recorded the validation loss of the last 100 epochs. In Figure 5 (a) and (b), we can see that the validation loss

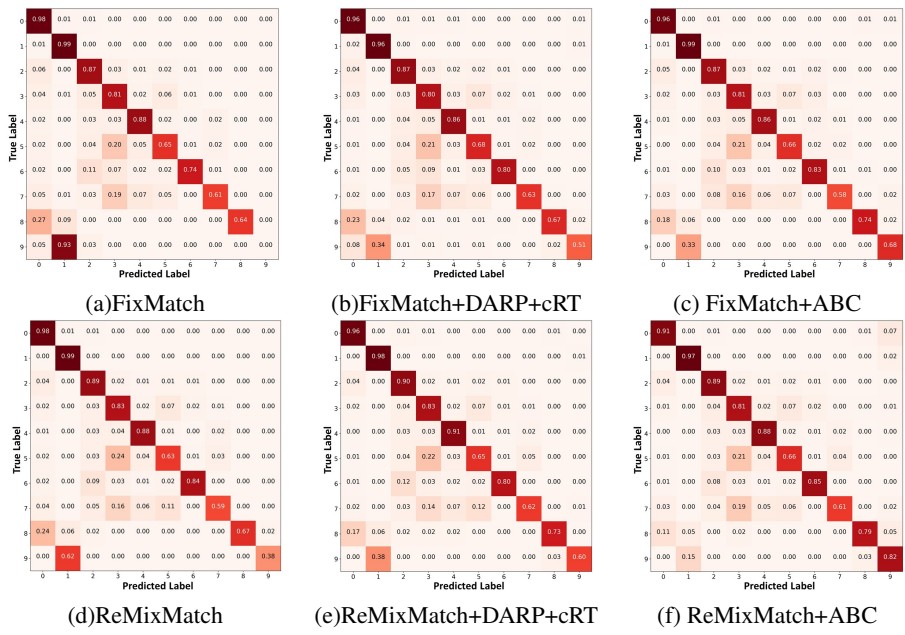

Figure 6: Confusion matrices of the predictions on the unlabeled data of CIFAR-10

of the algorithms based on decoupled learning of representations and a classifier tended to increase after a few epochs. The validation loss was reduced by conducting consistency regularization (C/R) using unlabeled data, but it still tended to increase. In the case of ReMixMatch+DARP+cRT+C/R* and FixMatch+DARP+cRT+C/R*, which do not fix the representations (algorithms marked with *), high-quality representations learned by the backbone were gradually replaced by the representations learned with a re-balanced classifier, which caused overfitting on minority classes. We can observe a similar tendency in Figure 5 (c) under the supervised learning setting. In contrast, the validation loss of ReMixMatch+ABC, FixMatch+ABC, and the proposed algorithm under supervised learning setting decreased steadily and achieved the lowest validation loss. The performances of the algorithms based on end-to-end training and decoupled learning of representations and a classifier are summarized in Table 9.

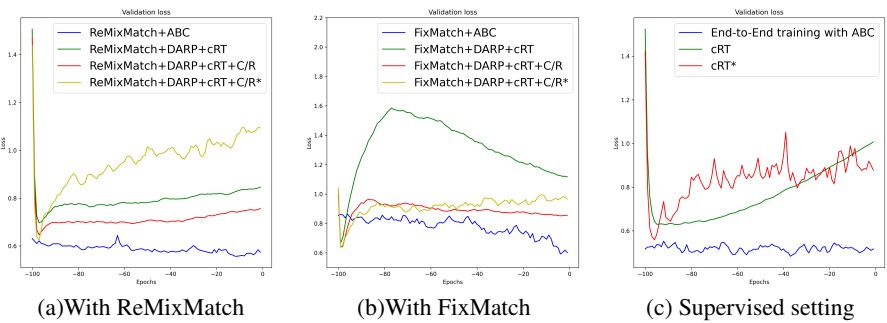

Figure 7: Validation loss graphs of algorithms based on end-to-end training and decoupled learning of representations and a classifier on the test set of CIFAR-10. The algorithms in (a) and (b) were trained on the train set of CIFAR-10-LT with $\gamma = 100$, $\beta = 20\%$, $N_1 = 1000$, and the algorithms in (c) were trained on the training set of CIFAR-10-LT with $\gamma = 100$, $\beta = 100\%$, $N_1 = 5000$. C/R and * in the graphs indicate consistency regularization and non-fixed representations, respectively.

Table 9: Performance of the algorithms based on end-to-end training and decoupled learning of representations and a classifier. The algorithms were trained on the training set described in the caption of Figure 7. The algorithms were tested on the test set of CIFAR-10.

| Performance of the algorithms based on end-to-end training versus decoupled learning | Overall | Minority |
|---|---|---|
| *Under the semi-supervised learning setting* | | |
| ReMixMatch+ABC (end-to-end training) | **82.4** | **75.7** |
| ReMixMatch+DARP+cRT+C/R* (Decoupled learning) | 80.6 | 71.4 |
| ReMixMatch+DARP+cRT+C/R (Decoupled learning) | 79.5 | 70.4 |
| ReMixMatch+DARP+cRT (Decoupled learning) | 78.5 | 66.4 |
| FixMatch+ABC (end-to-end training) | **81.1** | **72.0** |
| FixMatch+DARP+cRT+C/R* (Decoupled learning) | 80.3 | 71.6 |
| FixMatch+DARP+cRT+C/R (Decoupled learning) | 78.7 | 68.3 |
| FixMatch+DARP+cRT (Decoupled learning) | 78.1 | 66.6 |
| *Under the supervised learning setting* | | |
| End-to-End training of CNN with the ABC (end-to-end training) | **84.9** | **80.6** |
| cRT* (Decoupled learning of representations and the classifier of CNN) | 81.1 | 79.6 |
| cRT (Decoupled learning of representations and the classifier of CNN) | 80.0 | 79.9 |

## L  Ablation study for FixMatch [5] + ABC on CIFAR-10

Results in Table 10 show a similar tendency as that for ReMixMatch+ABC in Section 4.5 of the main paper.

Table 10: Ablation study for FixMatch+ABC on CIFAR-10-LT, $\gamma = 100$, $\beta = 20\%$

| Ablation study | Overall | Minority |
|---|---|---|
| FixMatch+ABC (proposed algorithm) | **81.1** | **72.0** |
| Without gradually decreasing the parameter of $\mathcal{B}(\cdot)$ for consistency regularization | 80.2 | 70.1 |
| Without consistency regularization for the ABC | 76.2 | 60.9 |
| Without using the 0/1 mask for the consistency regularization loss $L_{con}$ | 74.9 | 58.8 |
| Without using the 0/1 mask for the classification loss $L_{cls}$ | 77.1 | 62.7 |
| Without using the confidence threshold $\tau$ for consistency regularization | 79.2 | 67.8 |
| Using hard pseudo labels for consistency regularization | 78.8 | 68.0 |
| Without training backbone (ABC only) | 68.7 | 56.2 |
| Training the ABC with a re-weighting technique | 80.3 | 70.5 |
| Decoupled training of the backbone and ABC | 77.4 | 65.0 |