# OpenReview forum: "ABC: Auxiliary Balanced Classifier for Class-imbalanced Semi-supervised Learning"
_NeurIPS.cc/2021/Conference — NeurIPS 2021 Poster_

### Official Review · Reviewer_dcif · 2021-07-04

**Rating:** 7
**Confidence:** 5

**Summary:**

This paper deals with the class-imbalanced semi-supervised learning problem. The paper utilizes existing semi-supervised learning algorithms to acquire good representation, and additionally introduces an auxiliary balanced classifier called ABC, which is trained on a class-balanced subset of a minibatch to mitigate the biased results caused by class-imbalance. The ABC is trained using not only the balanced subset but also unlabeled data with consistency loss. The experimental results show that the proposed method outperforms existing methods by large margin.

**Limitations And Societal Impact:**

I believe the authors addressed these aspects in section 5.

**Main Review:**

The paper deals with a practically very important problem of class-imbalanced semi-supervised learning. I found the proposed method was reasonable and technically sound. Although the novelty of the method may not be outstanding given the prior works of [17], [3] and [4], the paper presents a good extension of these prior works to the practically important problem. The experiment is basically thorough and convincing. In addition, the paper is clearly and well written, thus easy to follow.

Overall, I think this is a good paper and has merit to the community, but I have some concerns and questions as follows.

* Is there any specific reason of using mask based on Bernoulli distribution to create the balanced subset? I think it is more straightforward to sample the same amount of data in each class to create balanced subset. For example, how about sampling data from uniform distribution using Gumbel softmax?
* According to Table 4, incorporating DARP seems harmful. Do authors have any thoughts on the reason of this phenomenon? Moreover, given this phenomenon, I think it is more reasonable to compare the proposed method with “w/ cRT” variant rather than “w/ DARP+cRT” variant in Table 1-3.
* L316-319: I am skeptical about the contribution of end-to-end training to the accuracy improvement. Looking at Table 5, the techniques related to consistency regularization, or using unlabeled data to train ABC, has dominant effect. For example, the accuracy drops to the same level as an existing method if the consistency regularization is removed; the variant of “Without using the 0/1 mask for the consistency regularization loss $L_{con}$” achieves overall accuracy of 79.0, which is close to 78.5 achieved by an existing CISSL method, namely ReMixMatch w/ DARP + cRT.

Minor comments and questions:
* L137-138: Is $\frac{N_{y_b}}{N_L}$ a mistake for $\frac{N_L}{N_{y_b}}$? The same doubt on L163-164, too.
* L163-164: How is $N_{\hat{q_b}}$ calculated? Is it calculated mini-batch wise?
* L252: “CRT” -> “cRT”
* L293: “ABC only (without backbone)”. Does this mean a single fully connected layer is attached directly on image input (I believe not)? The same question to L335-337, too. Please clarify the exact architecture in this condition.
* Table 5: Does each row mean the proposed algorithm + the condition in each row or does it mean all the conditions described earlier are also applied? For example, does “Without consistency regularization for the ABC” mean “Without gradually decreasing the parameter of B () for consistency regularization AND consistency regularization for the ABC”?
* [17] is published at ICLR 2020, not 2019


**Time Spent Reviewing:**

8 hours

---

> ### Author Response · Authors · 2021-08-10
> **Responses to your valuable comments**
>
> Thank you for your careful review of our paper and for the insightful and constructive comments. Please find our detailed answers to your comments below.
>
>
> ***<Responses to “some concerns and questions”>***
> **1. (Reason for using a mask to create a balanced subset)**
> If we sample a  balanced subset and train the ABC with that subset as you mentioned, we need to train the backbone and ABC using different minibatches each other, which will increase computational cost significantly compared to the training of the backbone alone. Therefore, instead of directly creating a balanced subset, we utilize a mask that leads to a balanced loss. This allows the backbone and ABC to be trained from the same minibatches, and thus the representations of minibatches calculated for training the backbone can be used for training the ABC once again. As a result, the ABC is possible to utilize high-quality representations learned by the backbone for its training (because the ABC is attached to the representation layer of the backbone) while only requiring slightly increased time cost compared to training the backbone alone, as we claim in Section 4.3 in the paper and Section G of the supplementary material.
>
> **2. (Decreased performance with DART in Table 4 & comparison with “w/cRT”)**
>  In Table 4 ([LSUN dataset [1]](https://arxiv.org/abs/1506.03365)), the decreased performance with the incorporation of [DARP [2]](https://papers.nips.cc/paper/2020/file/a7968b4339a1b85b7dbdb362dc44f9c4-Paper.pdf) is possibly because the scale of the dataset is very large. Specifically, DARP solves a convex optimization with the whole unlabeled data points to refine the pseudo labels. As the scale of the unlabeled dataset increases, this optimization problem becomes more difficult to solve, and consequently, the pseudo labels can be refined inaccurately.
> Following your suggestions, we conducted additional experiments using [FixMatch [3]](https://papers.nips.cc/paper/2020/hash/06964dce9addb1c5cb5d6e3d9838f733-Abstract.html)+[cRT [4]](https://openreview.net/forum?id=r1gRTCVFvB) and [ReMixMatch [5]](https://openreview.net/pdf/7e0bce0c7b750533163a2782f6af5b039305918c.pdf)+cRT, and compared their performance with that of FixMatch, ReMixMatch, FixMatch+DARP+cRT, ReMixMatch+DARP+cRT, and the proposed algorithm in Tables 1-3. The updated Tables 1-3 are as below. Incorporation of DARP increased the performance in most cases; but ReMixMatch+cRT performed better than ReMixMatch+DARP+cRT for some settings. In all cases in the updated Tables 1-3, the proposed algorithm performed best.
>
> **[Updated Table 1]**
>
> |                     |                           |                           |                          |
> |:---------------------:|:---------------------------:|:---------------------------:|:--------------------------:|
> |  Acc and Minor Acc  |        CIFAR-10-LT        |          SVHN-LT          |       CIFAR-100-LT       |
> |      Algorithm      | $\gamma$=100, $\beta$=20% | $\gamma$=100, $\beta$=20% | $\gamma$=20, $\beta$=40% |
> |       FixMatch      |         72.3, 53.8        |         88.0, 79.4        |        51.0, 32.8        |
> |     FixMatch+cRT    |         77.7, 64.7        |         90.0, 83.2        |        54.0, 42.6        |
> |  FixMatch+DARP+cRT  |         78.1, 66.6        |         89.9, 83.5        |        54.7, 41.2        |
> |     FixMatch+ABC    |        $81.1, 72.0$       |        $92.0, 87.9$       |       $56.3, 43.4$       |
> |      ReMixMatch     |         73.7, 55.9        |         89.8, 82.8        |        54.0, 37.1        |
> |    ReMixMatch+cRT   |         77.5, 62.7        |         92.1, 87.1        |        54.8, 43.1        |
> | ReMixMatch+DARP+cRT |         78.5, 66.4        |         92.1, 87.6        |        55.1, 43.6        |
> |    ReMixMatch+ABC   |        $82.4, 75.7$       |        $93.9, 92.5$       |       $57.6, 46.7$       |
>
> **[Update Table 2]**
>
> |                     |                           |                           |                          |                           |
> |:-------------------:|:-------------------------:|:-------------------------:|:------------------------:|:-------------------------:|
> |  Acc and Minor Acc  |                           |        CIFAR-10-LT        |                          |                           |
> |      Algorithm      | $\gamma$=100, $\beta$=10% | $\gamma$=100, $\beta$=30% | $\gamma$=50, $\beta$=20% | $\gamma$=150, $\beta$=20% |
> |       FixMatch      |         70.0, 48.9        |         74.9, 58.2        |        81.2, 70.7        |         68.5, 45.8        |
> |     FixMatch+cRT    |         72.8, 56.0        |         81.0, 71.6        |        83.3, 76.4        |         73.9, 57.9        |
> |  FixMatch+DARP+cRT  |         74.6, 59.2        |         79.0, 67.7        |        83.6, 77.1        |         73.2, 57.1        |
> |     FixMatch+ABC    |        $77.2, 65.7$       |        $81.5, 72.9$       |       $85.2, 80.2$       |        $77.1, 64.4$       |
> |      ReMixMatch     |         71.5, 52.2        |         75.8, 59.4        |        81.5, 70.7        |         69.9, 48.4        |
> |    ReMixMatch+cRT   |         75.9, 61.8        |         82.0, 73.0        |        85.5, 79.2        |         76.1, 60.9        |
> | ReMixMatch+DARP+cRT |         75.9, 62.1        |         81.0, 70.7        |        84.5, 77.8        |         73.9, 57.4        |
> |    ReMixMatch+ABC   |        $79.8, 70.8$       |        $84.3, 80.6$       |       $87.5, 84.6$       |        $80.6, 72.1$       |
>
> **[Updated Table 3]**
>
> |                   |                           |                           |                           |                           |
> |:-----------------:|:-------------------------:|:-------------------------:|:-------------------------:|:-------------------------:|
> |                   |                           |       CIFAR-10-STEP       | $\gamma$=100, $\beta$=20% |                           |
> | Acc and Minor Acc | $\gamma$=100, $\beta$=10% | $\gamma$=100, $\beta$=30% |  $\gamma$=50, $\beta$=20% | $\gamma$=150, $\beta$=20% |
> |     Algorithm     |            w/ -           |           w/ cRT          |        w/ DARP+cRT        |           w/ ABC          |
> |      FixMatch     |         54.0, 11.8        |         62.9, 30.7        |         69.8, 45.1        |        $75.9, 57.0$       |
> |     ReMixMatch    |         60.8, 25.1        |         70.0, 45.7        |         72.3, 50.6        |        $76.4, 65.7$       |
>
> **3. (Contribution of end-to-end training)**
> We agree that consistency regularization using unlabeled data had a more significant effect on performance improvement than end-to-end training. To emphasize this, we will switch the order of the two reasons for the performance improvement in L316-319. However, we still believe that end-to-end training can also contribute to improved performance, based on our experimental results in Figure 4 and Table 5 of the paper and Figure 7 of the supplementary material. For example, in our ablation study in Table 5, the proposed algorithm with "Decoupled training of the backbone and ABC" showed a decrease in the overall accuracy by 2.9%.
>
> Furthermore, to clarify the example that you mentioned, we also compare the performance of ReMixMatch+cRT and ReMixMatch+ABC “without consistency regularization for the ABC” (not “without using the 0/1 mask for the consistency regularization loss Lcon”). As can be seen from the updated Table 1 in our response to your comment #2, ReMixMatch+cRT achieved an overall accuracy of 77.5% on CIFAR-10-LT, which is 1.9% lower than ReMixMatch+ABC “without consistency regularization for the ABC.” These results show the potential advantage of end-to-end training.
>
> ***<Responses to “Minor comments and questions”>***
> -  L137-138: Yes, those are typos. We will correct these typos.
> -  L163-164:  $\hat{q_b}$ is the hard pseudo label of the $b$th unlabeled data point of a mini-batch, and $N_{\hat{q_b}}$ is the number of labeled data points (not unlabeled data points) that have the label $\hat{q_b}$ in the whole training set. Therefore, $N_{\hat{q_b}}$ is automatically determined from data once $\hat{q_b}$ is predicted.
> - L252: Sorry for the typo. We will correct CRT as cRT.
> - L293: “ABC only” means ABC-attached-[WideResNet 28-2 [6]](https://arxiv.org/abs/1605.07146) (CNN) that is trained only with the loss computed from ABC without using any other SSL algorithm for representation learning. We will clarify this in the revised version. Additionally, in order not to confuse the readers, we will change the expression “ABC only (without backbone)” to “ABC (without SSL backbone).”
> - Table 5:  Each row means the proposed algorithm+the condition in each row. For example  “Without gradually decreasing the parameter of B () for consistency regularization” means the proposed algorithm without gradually decreasing the parameter of B(), but with consistency regularization for the ABC. In order not to confuse the readers, we will clarify this in the revised version of our paper.
> - [17]: Sorry for the mistake. We will correct it as ICLR 2020.
>
> ***
> **Reference**
> [1] Yu et al. (2015). Lsun: Construction of a large-scale image dataset using deep learning with humans in the loop. arXiv preprint arXiv:1506.03365.
> [2] Kim et al. Distribution aligning refinery of pseudo-label for imbalanced semi-supervised learning. NeurIPS 2020.
> [3] Sohn et al. Fixmatch: Simplifying semi-supervised learning with consistency and confidence. NeurlPS 2020.
> [4] Kang et al. Decoupling representation and classifier for long-tailed recognition. ICLR  2020.
> [5] Berthelot et al. Remixmatch: Semi-supervised learning with distribution matching and augmentation anchoring. ICLR 2019.
> [6] Zagoruyko and Komodakis (2016). Wide residual networks. arXiv preprint arXiv:1605.07146.

---

> > ### Comment · Reviewer_dcif · 2021-08-11
> > **Thanks for the feedback**
> >
> > I appreciate the authors' feedback.
> > I have carefully read the feedback, comments from other reviewers, and the authors' response to them.
> > The explanation and additional results the authors provided successfully addressed my concerns.
> >
> > Once again, I think the paper is a good extension of the prior works.
> >
> > The method is technically sound and well motivated.
> > I believe this aspect has become even stronger with the authors' responses such as the one about the motivation behind the design of using binary mask and the benefit of end-to-end training.
> >
> > The experiment is thorough and the results are convincing.
> > This aspect has also become stronger with the additional results in authors' response to my comment and that to reviewer zh24.
> >
> > I believe the paper has clear merit to the community.
> > I am positive in accepting this paper.

---

> > > ### Author Response · Authors · 2021-08-11
> > > **Thanks for your response!**
> > >
> > > Thank you for your additional efforts and time to carefully read our response.
> > > We are happy to hear that our response successfully addressed your concerns.
> > > We sincerely appreciate your insightful comments that helped us clarify the novelties of our method and improve the manuscript.

---

### Official Review · Reviewer_zh24 · 2021-07-16

**Rating:** 6
**Confidence:** 4

**Summary:**

This work takes class-imbalanced data distribution into account for semi-supervised learning, which is prevailing in realistic scenarios. The problem setting is reasonable and is under-explored.


**Ethics Review Area:**

["I don’t know"]

**Main Review:**

The proposed algorithm, ABC, attaches an auxiliary balanced classifier to the representation layer of an existing SSL algorithm. The auxiliary classifier uses representations learned from all data points and is trained on a class-balanced subset so that it can avoid the bias towards the majority classes, thus improve generalization performance.

I have some concerns about this work:
a) This novelty is limited. As the authors point out, the superiority of ABC probably lies in that it is trained using unlabeled data, which is not adopted by previous methods. The performance improvement of ABC seems not surprising.
b) The failure of class-imbalanced learning (CIL) techniques (i.e., BALMS, LDAM) is obvious because they do not use unlabeled data for training. But what if we combine the existing CIL methods with SSL modules to make use of unlabeled data? I think it is a straightforward solution, and I wonder what is the superiority of ABC compared to such solutions.
c) The conclusion is too wordy. Some redundant explanations can be deleted to make it more concise.

>>

Thanks for the author response. It addresses my concerns on the superior performance for the proposal. I increase my score.


**Time Spent Reviewing:**

0.8

---

> ### Author Response · Authors · 2021-08-10
> **Responses to your valuable comments**
>
> Thank you for your careful review of our paper and for the insightful and constructive comments. Please find our detailed answers to your comments below.
>
>
> ***<Responses to “Main Review”>***
> **a) (Novelty)**
>  As you mentioned, the superior performance of the proposed algorithm over the existing class-imbalance learning (CIL) algorithm is mainly due to the use of unlabeled data. However, we would like to emphasize that it is not straightforward to make CIL algorithms utilize unlabeled data, because the existing CIL algorithms are designed for labeled data only. In our study, we propose a novel, well-designed, framework for utilizing unlabeled data while mitigating class imbalance. As you mentioned, the superior performance of the proposed algorithm over an existing CIL algorithm (which does not use unlabeled data) based on direct comparison seems not surprising. Here, we would like to clarify that the main purpose for the comparison is to demonstrate the benefits of utilizing unlabeled data in a well-designed CISSL algorithm, rather than to directly compare the performance.
>
> The way the ABC utilizes unlabeled data is novel in the following aspects.
> - First, the ABC utilizes unlabeled data in a “balanced” manner by using 0/1 mask that leads to a class-balanced loss. In contrast, the existing CIL algorithms may utilize unlabeled data toward the majority class even if they are combined with a SSL module.
> - Second, the ABC utilizes unlabeled data efficiently. Specifically, the ABC mitigates class imbalance by using 0/1 mask that has the same effect as creating a class-balanced set of a minibatch, instead of directly creating a balanced subset of training data. This allows the backbone and ABC to be trained from the same minibatches, and thus the representations learned by the backbone can be used for training the ABC once again. As a result, the ABC only requires slightly increased time cost compared to the cost for training the backbone alone, as we claim in Section 4.3 in the paper and Section G of the supplementary material.
> - Finally, we enabled end-to-end learning of representations and ABC so that ABC can be optimized to be fully compatible with representations that the backbone learns.Through experiments, we confirmed that training ABC and representations end-to-end produced better performance than [decoupled learning [1]](https://openreview.net/forum?id=r1gRTCVFvB) of representations and ABC, which is a recent trend of CIL. Therefore, our study demonstrates the potential advantage of end-to-end training as an alternative direction for CIL.
>
> **b) (Comparison with CIL techniques combined with a SSL module)**
> As you mentioned, it would be interesting to see the results of CIL techniques ([LDAM [2]](https://papers.nips.cc/paper/2019/hash/621461af90cadfdaf0e8d4cc25129f91-Abstract.html), [cRT [1]](https://openreview.net/forum?id=r1gRTCVFvB), [BALMS [3]](https://papers.nips.cc/paper/2020/hash/2ba61cc3a8f44143e1f2f13b2b729ab3-Abstract.html)) when they also utilize unlabeled data. However, as we answered to your comment (a) above, the existing CIL algorithms are designed for labeled data only, and thus it is not straightforward to combine them with a SSL module. For example, whereas [ReMixMatch [4]](https://openreview.net/pdf/7e0bce0c7b750533163a2782f6af5b039305918c.pdf) uses interpolations of data points as inputs, CIL techniques use each data point separately; therefore, if a CIL technique is incorporated into ReMixMatch directly, the predictions of the resulting algorithm can collapse into a specific class. Nevertheless, we performed additional experiments by combining three CIL techniques (LDAM, cRT, BALMS) with ReMixMatch in our own best way. Specifically, we combined a CIL technique with ReMixMatch in a way that representations are learned using ReMixMatch and then a classifier is fine-tuned with the learned representations (decoupled learning of representations and a classifier).
>
> The experimental results are as in the table below. We can see that although the combination of ReMixMatch significantly increased the performance of the CIL techniques, they still performed worse than the proposed algorithm. The superior performance of the proposed algorithm over the CIL techniques combined with ReMixMatch may be explained by the following reasons.
> - First, even though the CIL techniques combined with ReMixMatch also used unlabeled data for representation learning, these algorithms cannot use the unlabeled data in a “balanced” manner when tuning the classifier. In contrast, the proposed algorithm can use the unlabeled data in a “balanced” manner for training the ABC by conducting consistency regularization for ABC with 0/1 mask.
> - Second, as we answered to your comment (a) above, end-to-end training of the proposed algorithm possibly enabled better performance than decoupled learning of representations and a classifier. Using end-to-end training, ABC can be optimized to be fully compatible with the representations.
>
> |                   |                           |                           |                          |
> |:-----------------:|:-------------------------:|:-------------------------:|:------------------------:|
> | Acc and Minor Acc |        CIFAR-10-LT        |          SVHN-LT          |       CIFAR-100-LT       |
> |     Algorithm     | $\gamma$=100, $\beta$=20% | $\gamma$=100, $\beta$=20% | $\gamma$=20, $\beta$=40% |
> |        LDAM       |         61.7, 39.0        |         79.6, 64.7        |        46.7, 31.1        |
> |        cRT        |         66.4, 46.4        |         82.6, 70.4        |        48.5, 34.3        |
> |       BALMS       |         70.7, 69.8        |         87.6, 85.0        |        50.2, 42.9        |
> |  ReMixMatch+LDAM  |         74.2, 56,7        |         90.4, 84.0        |         54.1,38.3        |
> |   ReMixMatch+cRT  |         77.5,62.7         |         92.1,87.1         |         54.8,43.1        |
> |  ReMixMatch+BALMS |         79.9, 71.2        |         92.9, 89.2        |        56.4, 45.5        |
> |   ReMixMatch+ABC  |        $82.4, 75.7$       |        $93.9, 92.5$       |       $57.6, 46.7$       |
>
>
> **c) (Wordy conclusion)**
> We will delete some explanations and make the conclusion more concise in the revised version of our paper.
>
> ***<Responses to “Limitations And Societal Impact”>***
> Currently, we addressed these in Conclusion. To clarify, we will address these as a separate section.
>
> ***
> **Reference**
> [1] Kang, B., Xie, S., Rohrbach, M., Yan, Z., Gordo, A., Feng, J., and Kalantidis, Y. (2020). Decoupling representation and classifier for long-tailed recognition. In International Conference on Learning Representations.
> [2] Cao, K., Wei, C., Gaidon, A., Arechiga, N., and Ma, T. (2019). Learning imbalanced datasets with label-distribution-aware margin loss. In Wallach, H., Larochelle, H.,   Beygelzimer, A., d'Alché-Buc, F., Fox, E., and Garnett, R., editors, Advances in Neural Information Processing Systems, volume 32. Curran Associates, Inc.
> [3] Ren, J., Yu, C., sheng, s., Ma, X., Zhao, H., Yi, S., and Li, h. (2020). Balanced meta-softmax for long-tailed visual recognition. In Larochelle, H., Ranzato, M., Hadsell, R., Balcan, M. F., and Lin, H., editors, Advances in Ne
> [4] Berthelot, D., Carlini, N., Cubuk, E. D., Kurakin, A., Sohn, K., Zhang, H., and Raffel, C. (2019a). Remixmatch: Semi-supervised learning with distribution matching and augmentation anchoring. In International Conference on Learning Representations.

---

> ### Author Response · Authors · 2021-08-30
> **Thanks for your response!**
>
> Thank you for your additional efforts and time to carefully read our response. We are happy to hear that our response successfully addressed your concerns. We sincerely appreciate your insightful comments that helped us clarify the novelties of our method and improve the manuscript.

---

### Official Review · Reviewer_Jr6h · 2021-07-16

**Rating:** 6
**Confidence:** 2

**Summary:**

The paper proposes a new algorithm for semi-supervised learning in the class imbalanced setting by adding to the neural architecture an additional classifier head which is trained on the balanced datasets (through resampling).


**Limitations And Societal Impact:**

The authors measure the classification performance only with accuracy on a balanced datasets and minority accuracy. It would be interesting to see other metrics appropriate for imbalanced data such as G-mean as well.

**Main Review:**

Originality.
The idea of using random resampling inversely proportionally to class cardinality is rather standard. The architectures with an additional classification head as well as consistency regularization are also known, but their combination is quite interesting.

Quality.
The experimental results show that the proposed method improves the classification performance on three real datasets. The ablation study is performed.

Clarity.
The paper is in places hard to follow. The authors should add the pseudocode summarizing the proposed approach.  Some details that seem to have a non-neglectable impact on the results, like "gradually decreasing the parameter of B(·)for consistency regularization" are not strongly motivated and can be easily overlooked. The discussion of results is sometimes laconic.



**Time Spent Reviewing:**

4

---

> ### Author Response · Authors · 2021-08-10
> **Responses to your valuable comments**
>
> Thank you for your careful review of our paper and for the insightful and constructive comments. Please find our detailed answers to your comments below.
>
>
> ***<Responses to “Clarity”>***
> **1. (Pseudocode)**
> The pseudocode summarizing the proposed approach is provided in Section B of the supplementary material. We currently provide the pseudocode in the supplementary material due to page limit, but we will put it in the main paper if an additional page is allowed during a later phase.
>
> **2. (Explanations for gradually decreasing the parameter for B(·))**
> Let us explain the details of gradually decreasing the parameter of  B(·) for consistency regularization. In the early stage of training, the confidence for prediction of unlabeled data is low, and thus only few unlabeled data points have prediction confidence higher than the confidence threshold $\tau$. To take full advantage of these few data points for consistency regularization, we set the parameter of B(·) to be close to 1 in the early stage of training. As the training continues and the number of unlabeled data points with prediction confidence higher than $\tau$ increases, we gradually decrease the Bernoulli distribution parameter and finally set it to $N_L/N_{\hat{q}_b}$ to obtain a balanced classifier. In the revised version of our paper, we will add this description and fill in the missing details.
>
> **3. (Sometimes laconic discussion of results)**
> The discussion of results sometimes laconic is mainly due to page limit. In such cases, we have tried to provide more details in the supplementary material. We will revise our manuscript by adding more detailed discussions, particularly those for addressing the comments from the reviewers.
>
> ***<Responses to “Limitations And Societal Impact”>***
> **(More metrics such as G-mean)**
> Following your suggestion, we performed additional experiments on [CIFAR-10-LT, CIFAR-100-LT [1]](http://www.cs.toronto.edu/~kriz/learning-features-2009-TR.pdf) and [SVHN [2]](https://research.google/pubs/pub37648/) for the main setting of the paper and measured the  performance of the competing algorithms using an additional metric, the geometric mean (G-Mean) of class-wise accuracy. As can be seen from the following table, the proposed algorithm recorded the highest G-Mean in all settings, indicating that the proposed algorithm performs effectively in a balanced way for all classes. In the case of the experiment on CIFAR-100, some algorithms produced 0 % accuracy for a specific class and resulted in 0% of G-Mean.
>
> In addition to G-Mean, we also measured the standard deviation of class-wise accuracy (STDC) and the difference between the maximum class-wise accuracy and minimum class-wise accuracy (MaxMin). A smaller value of each metric indicates better performance.  As can be seen from the table below, the proposed algorithm recorded the lowest values of the metrics in most settings. Although [FixMatch [3]](https://papers.nips.cc/paper/2020/hash/06964dce9addb1c5cb5d6e3d9838f733-Abstract.html)+ABC produced higher STDC and MaxMin than FixMatch+[DARP [4]](https://papers.nips.cc/paper/2020/file/a7968b4339a1b85b7dbdb362dc44f9c4-Paper.pdf)+[cRT [5]](https://openreview.net/forum?id=r1gRTCVFvB)  on CIFAR-100-LT, when [ReMixMatch [6]](https://openreview.net/pdf/7e0bce0c7b750533163a2782f6af5b039305918c.pdf) was used instead of FixMatch as the backbone, ReMixMatch+ABC showed lower STDC and MaxMin than ReMixMatch+DARP+cRT on the same dataset.
>
> **[Performance comparison using G-mean]**
>
> |                 |                           |                           |                          |
> |:-----------------:|:---------------------------:|:---------------------------:|:--------------------------:|
> | G-Mean  |  CIFAR-10-LT |    SVHN-LT   | CIFAR-100-LT |
> |         Algorithm        | $\gamma$=100, $\beta$=20% | $\gamma$=100, $\beta$=20% |  $\gamma$=20, $\beta$=40% |
> |         FixMatch         |     62.0     |     87.3     |       0      |
> |       FixMatch+CReST+PDA       |     74.4     |     88.6     |     42.3     |
> |          FixMatch+DARP         |     71.5     |     87.6     |       0      |
> |        FixMatch+DARP+cRT       |     76.7     |     89.8     |     47.0     |
> |          FixMatch+ABC          |     $80.5$     |     $91.8$    |     $49.0$     |
> |        ReMixMatch        |     62.5     |     89.5     |       0      |
> |       ReMixMatch+CReST+PDA       |     72.2     |     90.7     |       0      |
> |          ReMixMatch+DARP         |     71.9     |     89.7     |     42.5     |
> |        ReMixMatch+DARP+cRT       |     77.9     |     92.0     |     48.3     |
> |          ReMixMatch+ABC          |     $81.9$    |     $93.8$     |     $50.8$     |
>
> **[Performance comparison using STDC and MaxMin]**
>
> |                 |                           |                           |                          |
> |:-----------------:|:---------------------------:|:---------------------------:|:--------------------------:|
> | STDC and MaxMin |        CIFAR-10-LT        |          SVHN-LT          |       CIFAR-100-LT       |
> |    Algorithm    | $\gamma$=100, $\beta$=20% | $\gamma$=100, $\beta$=20% | $\gamma$=20, $\beta$=40% |
> |     FixMatch    |         25.9, 92.5        |         11.2, 37.8        |        28.6, 96.0        |
> |   FixMatch+CReST+PDA  |         17.5, 56.3        |         9.7, 31.8         |        24.9, 94.0        |
> |     FixMatch+DARP     |         19.3, 60.1        |         10.8, 35.4        |        27.8, 93.0        |
> |   FixMatch+DARP+cRT   |         13.6, 42.4        |         7.7, 28.4         |       $22.7, 81.0$       |
> |      FixMatch+ABC     |        $11.5, 34.0$       |        $5.7, 17.5$        |        24.0, 86.0        |
> |    ReMixMatch   |         25.3, 89.6        |         9.4, 31.9         |        27.7, 99.0        |
> |   ReMixMatch+CReST+PDA  |         19.4, 65.5        |         7.3, 25.1         |        28.1, 95.0        |
> |     ReMixMatch+DARP     |         19.7, 62.9        |         8.7, 29.6         |        26.2, 95.0        |
> |   ReMixMatch+DARP+cRT   |         13.9, 41.5        |         5.4, 17.8         |        23.3, 87.0        |
> |      ReMixMatch+ABC     |        $10.4, 32.6$       |        $3.3, 10.5$        |       $21.5, 86.0$       |
>
>
> ***
> **Reference**
> [1] Krizhevsky, A. (2009). Learning multiple layers of features from tiny images. Technical report, Department of Computer Science, University of Toronto.
> [2] Netzer, Y., Wang, T., Coates, A., Bissacco, A., Wu, B., and Ng, A. Y. (2011). Reading digits in natural images with unsupervised feature learning. In NIPS Workshop, 2011.
> [3] Sohn, K., Berthelot, D., Carlini, N., Zhang, Z., Zhang, H., Raffel, C. A., Cubuk, E. D., Kurakin, A., and Li, C.-L. (2020). Fixmatch: Simplifying semi-supervised learning with consistency and confidence. Advances in Neural Information Processing Systems, 33.
> [4] Kim, J., Hur, Y., Park, S., Yang, E., Hwang, S. J., and Shin, J. (2020a). Distribution aligning refinery of pseudo-label for imbalanced semi-supervised learning. In Larochelle, H., Ranzato, M., Hadsell, R., Balcan, M. F., and Lin, H., editors, Advances in Neural Information Processing Systems, volume 33, pages 14567–14579. Curran Associates, Inc.
> [5] Kang, B., Xie, S., Rohrbach, M., Yan, Z., Gordo, A., Feng, J., and Kalantidis, Y. (2020). Decoupling representation and classifier for long-tailed recognition. In International Conference on Learning Representations.
> [6] Berthelot, D., Carlini, N., Cubuk, E. D., Kurakin, A., Sohn, K., Zhang, H., and Raffel, C. (2019a). Remixmatch: Semi-supervised learning with distribution matching and augmentation anchoring. In International Conference on Learning Representations.

---

> > ### Comment · Reviewer_Jr6h · 2021-08-18
> > **Thank you for the responses**
> >
> > Thank you for the responses! I think that the paper is interesting, and the provided additional results made me more confident about the approach's usefulness.
> >
> > P.S. in multi-class imbalanced learning, we often use G-mean with correction to avoid zeroing, see e.g. imbalanced-learn python package docs.

---

> > > ### Author Response · Authors · 2021-08-19
> > > **Updated table with G-mean with correction to avoid zeroing**
> > >
> > > Thank you for your additional efforts and time to carefully read our response. We are glad to hear that our response made you more confident about the usefulness of our approach.
> > >
> > > Following your comment, we conducted additional experiments using G-mean with correction to avoid zeroing, for the dataset CIFAR-100-LT for which the G-mean without correction previously returned zero. We set the parameter for correction as 1%, which indicates that the minimum class-wise accuracy is 1%. The updated table is as follows.
> > >
> > > We will add this version of table in the final version of our paper. Thank you for helping us further improve our paper!
> > >
> > > **[Performance comparison using G-mean with correction to avoid zeroing]**
> > >
> > > |                 |                           |                           |                          |
> > > |:-----------------:|:---------------------------:|:---------------------------:|:--------------------------:|
> > > | G-Mean  |  CIFAR-10-LT |    SVHN-LT   | CIFAR-100-LT |
> > > |         Algorithm        | $\gamma$=100, $\beta$=20% | $\gamma$=100, $\beta$=20% |  $\gamma$=20, $\beta$=40% |
> > > |         FixMatch         |     62.0     |     87.3     |       38.5      |
> > > |       FixMatch+CReST+PDA       |     74.4     |     88.6     |     42.3     |
> > > |          FixMatch+DARP         |     71.5     |     87.6     |       40.4      |
> > > |        FixMatch+DARP+cRT       |     76.7     |     89.8     |     47.0     |
> > > |          FixMatch+ABC          |     $80.5$     |     $91.8$    |     $49.0$     |
> > > |        ReMixMatch        |     62.5     |     89.5     |       41.2      |
> > > |       ReMixMatch+CReST+PDA       |     72.2     |     90.7     |       43.1      |
> > > |          ReMixMatch+DARP         |     71.9     |     89.7     |     42.5     |
> > > |        ReMixMatch+DARP+cRT       |     77.9     |     92.0     |     48.3     |
> > > |          ReMixMatch+ABC          |     $81.9$    |     $93.8$     |     $50.8$     |

---

### Official Review · Reviewer_2qyx · 2021-07-17

**Rating:** 6
**Confidence:** 4

**Summary:**

The authors introduce a semi-supervised learning algorithm (SSL) for class-imbalanced scenarios.  In particular, an Auxiliar Balanced Classifier coupled with a 0/1 mask is used to favor the cost regularization. The methodology is interesting, and the experiments demonstrate that the proposed method outperforms state-of-the-art techniques on image classification tasks, even for large datasets under challenging imbalance ratios.

**Limitations And Societal Impact:**

The proposed method is helpful for several artificial intelligence applications, in particular, scenarios including semi-supervised and imbalanced datasets. Then, societal impact, though not directly referenced, is clear concerning the presented approach.

**Main Review:**

Though the paper does not present an outstanding theoretical advance, the methodology seems to benefit real-world tasks for semi-supervised and imbalanced datasets. Overall, the manuscript is easy to follow, holding careful experimentation and method comparison.

Pros.

-A consistency regularization approach is coupled within a deep learning framework for image classification under unlabelled data.

-In general, deep learning models treat class-imbalanced-datasets problems only from labeled data; this algorithm uses unlabelled data during training.

-The experiments include several scenarios (easy and complex), demonstrating that the author's proposal outperforms the included literature.

-An ablation analysis is presented, describing in detail the role of each of the algorithm stages.

Cons. and comments

-The mathematical foundation could be enhanced. In particular, the threshold hyperparameter tau seems to be crucial for the mask.
However, neither experimental, not theoretical insights are provided.

-Please number all the equations in the manuscript.

-Please take care with the counters notations in lines 89-92. The authors mix i, n, and m indexes.


**Time Spent Reviewing:**

2

---

> ### Author Response · Authors · 2021-08-10
> **Responses to your valuable comments**
>
> Thank you for your careful review of our paper and for the insightful and constructive comments. Please find our detailed answers to your comments below.
>
>
> ***<Responses to “Cons. and comments”>***
> **1. (Specification of the threshold hyperparameter $\tau$)**
> In general, the confidence threshold $\tau$ should be set high enough, but not too high. If $\tau$ is low, training becomes unstable because many misclassified unlabeled data points would be used for training. However, if $\tau$ is too high, most of the unlabeled data points would not be used for consistency regularization. Based on these insights, we set $\tau$ as 0.95 in our experiments. We confirmed via experiments that this value of $\tau$ enabled high accuracy as well as stability. Specifically, we conducted experiments on [CIFAR-10-LT [1]](http://www.cs.toronto.edu/~kriz/learning-features-2009-TR.pdf) for the main setting while changing the value of $\tau$. We measured the validation accuracy of [ReMixMatch [2]](https://openreview.net/pdf/7e0bce0c7b750533163a2782f6af5b039305918c.pdf)+ABC during the last 50 epochs (1 epoch=500 iterations) of training and calculated the mean and standard deviation (STD) of these values. As can be seen from the table below, the proposed algorithm achieved the highest mean and lowest STD of the validation accuracy when $\tau$ was 0.95. When $\tau$ was set higher or lower than 0.95, the mean of the validation accuracy decreased. In particular, as the value of $\tau$ decreased from 0.95, the STD  increased rapidly, indicating instability of the training. In the revised version of our paper, we will add these discussions.
>
>
> |                 |            |            |             |               |             |            |            |            |
> |:---------------:|:----------:|:----------:|:-----------:|:-------------:|:-----------:|:----------:|:----------:|:----------:|
> |  ReMixMatch+ABC |            |            | CIFAR-10-LT | $\gamma$=100, | $\beta$=20% |            |            |            |
> |       $\tau$        |      1     |    0.98    |    $0.95$   |      0.9      |     0.85    |     0.8    |    0.75    |     0.7    |
> | Mean and STD | 78.9, 0.36 | 81.8, 0.34 | $82.3, 0.2$ |  81.3, 0.32   |  81.5, 0.39 | 81.2, 0.63 | 80.0, 2.87 | 79.0, 5.76 |
>
>
> **2. (Numbering of equations)**
> We will number all the equations in the revised version of our paper.
>
> **3. (Counters notations)**
> To avoid confusion, we will change $x_i$ and $y_i $ to $ x_n$ and $y_n$, respectively, and $u_i$ to $u_m$.
>
> ***
> **Reference**
>
> [1] Krizhevsky, A. (2009). Learning multiple layers of features from tiny images. Technical report, Department of Computer Science, University of Toronto.
> [2] Berthelot, D., Carlini, N., Cubuk, E. D., Kurakin, A., Sohn, K., Zhang, H., and Raffel, C. (2019a). Remixmatch: Semi-supervised learning with distribution matching and augmentation anchoring. In International Conference on Learning Representations.

---

> > ### Comment · Reviewer_2qyx · 2021-08-28
> > **Authors rebbutal discussion**
> >
> > Thank you for your answers. In particular, thank you for clarifying the tau influence experimentally. I suggest accepting the paper.

---

> > > ### Author Response · Authors · 2021-08-29
> > > **Thanks for your response!**
> > >
> > > Thank you for your additional efforts and time to carefully read our response. We are glad to hear that our response successfully addressed your concerns. Thank you for your comments that helped us improve our paper!

---

### Decision · Program_Chairs · 2021-09-27

**Decision:**

Accept (Poster)

**Comment:**

This paper presents a method for

This paper has four favorable reviews, describing beneficial methodology for semi-supervised learning for imbalanced data, and good experimental results for real data sets. In contrast, a reviewer pointed out that the combination of the existing class-imbalance learning (CIL) and semi-supervised learning is straightforward, and therefore the novelty is limited. The authors argue that it is not straightforward to utilize unlabeled data in CIL and that there are no such studies in the past. However, related research is not entirely absent; for example,
T. Iwata, A. Fujino, N. Ueda, "Semi-supervised Learning for Maximizing the Partial AUC," proc. of AAAI 2020 has already been proposed.
The authors should compare the proposed algorithm not only with CIL and naive combined methods, but also with other existing related works.